# Potential for aerobic hydrocarbon oxidation in archaea

Andy O. Leu ✉, Ben J. Woodcroft, Simon J. McIlroy & Gene W. Tyson ✉

Over the last decade, there have been significant advances in our understanding of anaerobic hydrocarbon oxidation in archaea. However, the ability to oxidise hydrocarbons aerobically has been described in bacteria but not yet in archaea. Here, we provide evidence supporting potential aerobic hydrocarbon oxidation ability in archaea belonging to a novel order within the class Syntropharchaeia, which we propose to name Candidatus 'Aerarchaeales'. This order is represented by six metagenome-assembled genomes (MAGs) spanning three genera that are found in terrestrial and marine ecosystems. In particular, MAGs belonging to a newly defined genus, Ca. 'Aerovita', encode a copper monooxygenase complex with homology to bacterial hydrocarbon monooxygenases. The presence of genes encoding other oxygen-dependent enzymes, such as haem-copper oxygen reductase, indicates that Ca. 'Aerovita' may be capable of aerobic respiration. Our findings suggest that horizontal gene transfer between archaeal and bacterial domains facilitated the evolution of aerobic hydrocarbon-oxidizing archaea.

Hydrocarbons are ubiquitous in the natural environment and are formed by both abiotic and biotic mechanisms[1,2]. However, global demand for hydrocarbons, primarily in the form of oil and gas for civil and industrial purposes continues to rise, leading to frequent contamination of water and soil, with estimates of 1.3 million litres of crude oil entering natural environments each year[3]. Additionally, large quantities of gaseous short-chain hydrocarbons are released into the atmosphere annually, where they contribute directly to the greenhouse effect or are abiotically converted into methane, a far more potent greenhouse gas[4,5]. Elucidating the microbial drivers of hydrocarbon oxidation is essential to predict their fate in the environment. However, our understanding of the diversity of microbial lineages involved in hydrocarbon oxidation remains incomplete.

In aerobic environments, oxidation of hydrocarbons is mediated by hydrocarbon monooxygenase (HMO), which belongs to the copper-containing membrane-bound monooxygenase superfamily (CuMMO). Monooxygenases belonging to this superfamily have been shown to oxidise methane (pMMO), ammonia (AMO), and hydrocarbons (HMO). While the pMMO are found exclusively in bacteria[6–8], AMOs have been found in several bacterial phyla and the archaeal lineage Nitrososphaeria[7,8]. The HMOs were initially thought to be confined to the Actinobacteria[9,10], but have recently been discovered in many phylogenetically divergent phyla including SAR324[11], Methylomirabilota[12] (previously Rokubacteria) and Desulfobacterota_B[13] (previously Binatota). These findings suggest that there may be other undiscovered HMO-encoding lineages.

While there has been no HMOs found in archaea, multiple archaeal lineages have been shown to anaerobically oxidise various hydrocarbons using the alkyl-coenzyme M reductase (ACR), a homologue of the methyl-coenzyme M reductase (MCR) enzyme, a key complex used by methanogens and anaerobic methanotrophs. This includes members of the Syntropharchaeia, such as Candidatus Syntropharchaeum[14] that can anaerobically oxidise propane and butane, and Candidatus Alkanophaga[15] that can oxidise different petroleum n-alkanes. In addition, two genera within Methanosarcinales, Candidatus Argoarchaeum[16] and Candidatus Ethanoperedens[17], were found to oxidise ethane. More recently, members of the class Methanoliparia were found to consume a variety of long-chain alkanes, n-alkylcyclohexanes, and n-alkylbenzenes using ACR[18]. Metagenomic studies have also revealed the presence of ACR in Methanomethylicia[19], Methanomassiliicoccales[19], Archaeoglobi[20], Bathyarchaeia[21], Helarchaeles[22] and Hadarchaeota[22], indicating that

Centre for Microbiome Research, School of Biomedical Sciences, Queensland University of Technology (QUT), Translational Research Institute, Woolloongabba, QLD, Australia. ✉e-mail: leua@qut.edu.au; gene.tyson@qut.edu.au

archaeal hydrocarbon oxidation is likely more phylogenetically and metabolically diverse than previously thought.

Here, we report the potential for aerobic hydrocarbon metabolism within a novel archaeal order of the Syntropharchaeia. Comparative genomics and metabolic reconstruction of six metagenome assembled genomes reveal the acquisition of the HMO complex and electron transport complexes within a novel genus, supporting their potential transition from a primarily anaerobic mixotrophic metabolism that includes organic matter degradation and $CO_2$ fixation towards an aerobic hydrocarbon-oxidising metabolism. This study presents the first evidence for the potential of aerobic hydrocarbon oxidation in Archaea using a copper monooxygenase-based mechanism that was hitherto considered exclusive to Bacteria.

## Results

### Recovery of a novel Syntropharchaeia clade

Given the recent discovery of anaerobic methane and hydrocarbon oxidation in Syntropharchaeia, a survey was performed to recover additional metagenome assembled genomes (MAGs) from this lineage. Using SingleM[23], we detected novel single-copy marker genes related to members of the Syntropharchaeia from 28 metagenomes where most were at low abundance (≥0.37–8.82X coverage). Coassembly of five metagenomes from waters collected at a Brazillian copper mine[24] provided sufficient coverage for the recovery of a single MAG, CM-1, that corresponded to the Syntropharchaeia detected in the survey. Using this MAG, we were able to refine our search (see Methods) leading to the recovery of additional closely related MAGs, CG-1 and CG-2, from Californian agriculture groundwater[25], MW-1 from groundwater aquifer samples in Mallorca island[26], and YL-1 and YL-2 from sediment samples from Yellowstone Lake (Supplementary Data 1). Based on CheckM2[27], these genomes are of mid to high completeness (79-94%) and low contamination (0.4–2.5%) with the exception of MW-1 (compl. = 28% and cont. = 0.07%). The extrapolated genome sizes range from 2.24 to 2.57 Mbp.

Comparative genomic analysis revealed the CG-2 and YL-2 MAGs were higher-quality versions of GCA_016207465.125[25] and spire_mag_01109967 from the SPIRE database[28], while CM-1, CG-1, MW-1, and YL-1 represented novel MAGs. A genome tree placed the six MAGs in a monophyletic clade within the class Syntropharchaeia (Fig. 1A). Based on GTDB-Tk[29], these MAGs were placed in the provisional order JACQPP01. Relative evolutionary divergence (RED) and amino acid-identity (AAI) indicate these genomes comprises two novel families and three genera within JACQPP01. The CM-1, CG-1, and MW-1 form a single genus (g_CM-1) within the same family (f_CG-2) as CG-2. The YL-1 and YL-2 MAGs represent the other novel family (f_YL-1). Average nucleotide identity (ANI) analysis revealed five distinct species (≤95% ANI), with YL-1 and YL-2 representing distinct strains of the same species (98.3% ANI).

Consistent with the genome tree, 16S rRNA gene-based phylogenetic analyses placed the JACQPP01 order as a novel lineage within the Euryarchaeota, closely related to ANME-1 and *Syntropharchaeaceae* (Fig. 1B). The 16S rRNA gene sequences from CM-1, CG-1, and MW-1 cluster (≥95% similarity) with a cloned environmental sequence from the Bor Kleung hot spring, Thailand[30]. The YL-1 and YL-2 cluster (≥95% similarity) with a sequence recovered from a sinkhole biomat in Cenote La Palita, Mexico, and are distantly related (≥89.05% similarity) to sequences from diverse environments, including hot springs in Greece[31], Taiwanese seafloor sediments, and Japanese groundwater[32]. These environmental sequences further reveal the broad ecological and geographical distribution for this lineage.

### Divergent HMO complex identified within novel Syntropharchaeia MAGs

To investigate the metabolic potential of these novel Syntropharchaeia species, metabolic reconstruction was performed on each of the MAGs. Surprisingly, genes encoding the methyl-coenzyme reductase (MCR) and alkyl-coenzyme reductase (ACR) complex could not be identified in members of JACQPP01. However, novel copper monooxygenase (CuMMO) complexes were found within the genomes

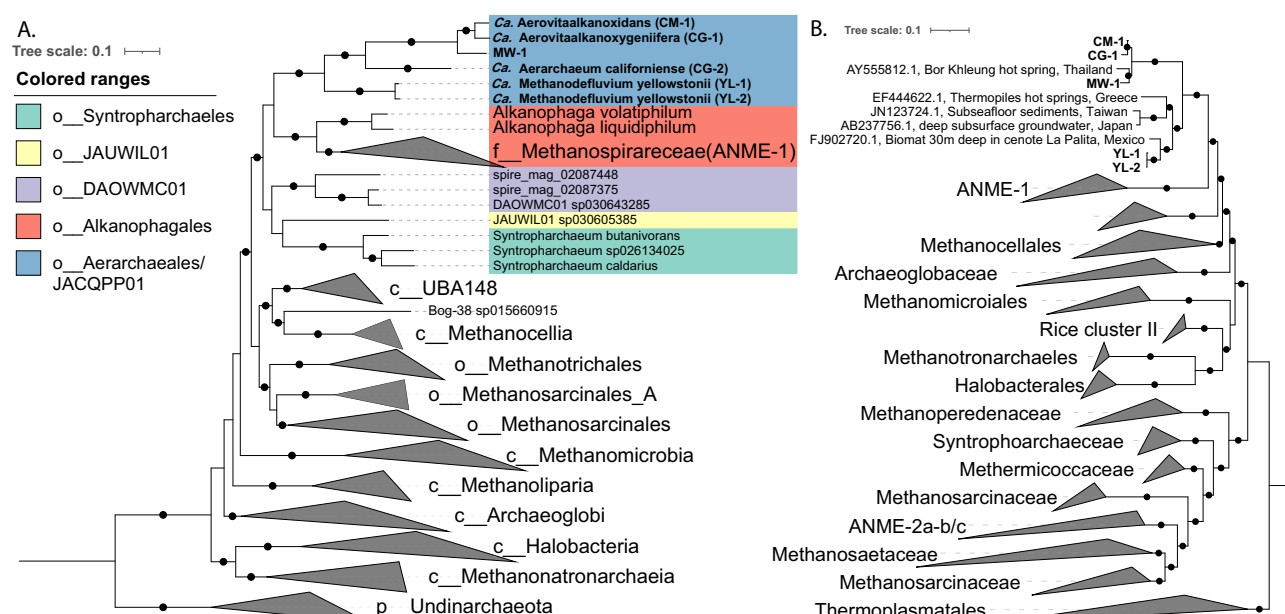

**Fig. 1 | Phylogenetic placement of the *Ca.* 'Aerarchaeales' MAGs. A** Maximum-likelihood tree inferred using GTDB-Tk using 52 single copy genes from 5,869 archaeal genomes. The tree was rooted using a Undinarchaeota outgroup. Genomes not recovered here were dereplicated to the best species per genera based on CheckM2. Genomes were coloured based on their order level placement, Blue: *Ca.* 'Aerarchaeales'; Red: Alkanophagales; Purple: DAOWMC01; Yellow: JAUWIL01;

Green; Syntropharchaeales. **B** Maximum-likelihood tree inferred from 16S rRNA sequences from *Ca.* 'Aerarchaeales' MAGs and the SILVA database v138.1. Bootstrap values were determined by non-parametric bootstrapping of 100 replicates. Black dots indicate ≤75% bootstrap support. The scale bar represents the number of amino acid and nucleotide substitutions per site, respectively. Bold labels denote the *Ca.* 'Aerarchaeales' MAGs recovered in this study.

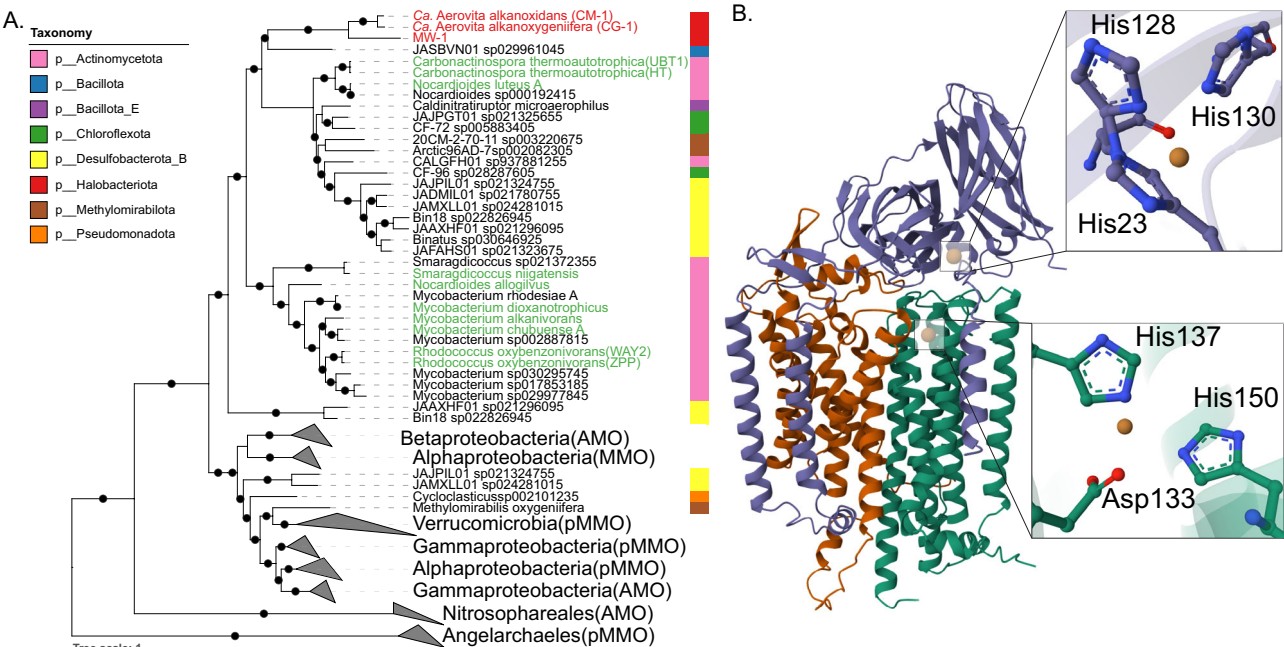

**Fig. 2 | Predicted hydrocarbon oxidation capacity of *Ca.* 'Aerovita' CuMMO.**
**A** Maximum-likelihood phylogenetic tree of CuMMO enzymes based on concatenated subunits A, B, and C. Clades are collapsed according to taxonomic class and known substrates in each clade are indicated in parenthesis. Genes indicated in red and green font denote the *Ca.* 'Aerovita' and verified HMO complexes oxidizing hydrocarbon, respectively. Bootstrap values were determined by non-parametric bootstrapping of 100 replicates. Black dots indicate ≥75% bootstrap support and the scale bar represent the number of amino acid substitutions per site. **B** Predicted HMO trimer complex 3D structure of CM-1. HMO subunits A, B, and C are coloured in red, purple, and green. Predicted copper binding sites based on alphafill are shown in focus with the copper ions depicted as orange-coloured spheres. See Supplementary Data 2 for the quality metrics of transplanted ligands.

belonging to g_CM-1 (Supplementary Fig. 1). The g_CM-1 CuMMO subunits A, B, and C had higher average amino acid identities to bacterial homologues (up to 50.8%, 41.1%, and 43.8%, respectively) than to archaeal homologues (up to 27.35%, 30.4%, and 34%, respectively). The MAGs from g_CG-2 and f_YL-1 did not encode genes for the CuMMO complexes.

To infer the putative substrate usage of the g_CM-1 CuMMO complexes and their evolutionary relationship to characterised CuMMO, a phylogenetic analysis based on a concatenated HmoABC/PmoABC/AmoABC alignment was performed. Our results showed that the CuMMO complexes from CM-1, CG-1, and MW-1 form a clade within the hydrocarbon monooxygenase (HMO) clade, distinct from the archaeal ammonia monooxygenases (AMOs) (Fig. 2A; Supplementary Fig. 2). Characterized HMOs have been shown to act specifically on short-chain hydrocarbons, including alkanes ($C_2$–$C_4$), alkenes, and chlorinated alkanes[33]. Thus, while the exact substrate(s) of the g_CM1 HMO proteins remain unknown, they are likely involved in the oxidation of short-chain hydrocarbons.

In addition to our novel archaeal HMO proteins, we identified additional HMO homologs in MAGs from diverse bacterial lineages that have not been previously reported, including the phyla Bacillota, Bacillota_E, and Chloroflexota. Notably, our HMO complexes clustered most closely with a MAG from a novel bacterial genus (JASBVN01) within the *Kyrpidiaceae* family (phylum Bacillota), which was recovered from coal-fire vents[34]. The sporadic distribution of HMO complexes across divergent phyla further supports horizontal gene transfer (HGT) as a major driver of the HMO evolutionary history. Based on our phylogenetic analysis, we propose that the g_CM1 HMO proteins were horizontally transferred from bacterial donors.

### Structural characterisation of the hmo complex
To verify that the HMO proteins are functionally conserved CuMMOs, we structurally predict the CM-1 and CG-1 HMO protein complexes using Colabfold[35], which predicted a well-folded 3D structure with high confidence (Fig. 2B and Supplementary Fig. 3A, B). Comparison of the predicted 3D structures using foldseek[36] revealed significant similarity to the crystal structure to the CuMMO from the proteobacterial methanotroph *Methylococcus capsulatus* (PDB structure 7T4P44; Probability ≥0.99, qTM = 0.9, tTM = 0.31). Ligand binding predictions showed the complexes also contained two conserved metal-binding sites common to CuMMO complexes[37,38] (Fig. 2B; Supplementary Data 2). In the HmoB subunits, all three histidine residues required for the mononuclear copper site (CuB) were found (Fig. 2B; Supplementary Fig. 4A). Similarly, the HmoC subunits showed conservation of the aspartic acid and histidine residues required for the copper site (CuC) (Supplementary Fig. 4B). Together with the phylogenetic analysis, the highly resolved structures with conserved metal-binding sites suggest that the g_CG-1 CuMMO complexes are functionally conserved hydrocarbon monooxygenases.

### Emergence of aerobic metabolism in family CG-2
Unlike other anaerobic hydrocarbon-oxidizing Syntropharchaeia, the presence of the HMO complex in members of g_CM-1 suggests they are capable of aerobic respiration. However, the non-HMO encoding MAG CG-2, within the same family (f_CG-2), also appears to be capable of aerobic respiration. These MAGs did not encode oxygen-sensitive protein complexes found in other Syntropharchaeia, such as the CODH/ACS complexes or most of the methanogenesis pathway. Instead, all MAGs within f_CG-2 encode genes for haem-copper oxygen reductase (complex IV; CoxABCD) and the haem A and O synthase (CtaA and CtaB), supporting their use of oxygen for respiration[39–42] (Fig. 3).

While the substrate range of CuMMO enzymes can be broad, with pMMOs capable of oxidizing methane, short-chain alkanes ($C_2$–$C_5$), alkenes ($C_2$–$C_4$)[43,44], and acetone[45], and AMOs known to act on ammonia, as well as alkanes (≤$C_8$) and alkenes (≤$C_5$)[46], the actual substrate specificity is often determined by downstream metabolic pathways. Key

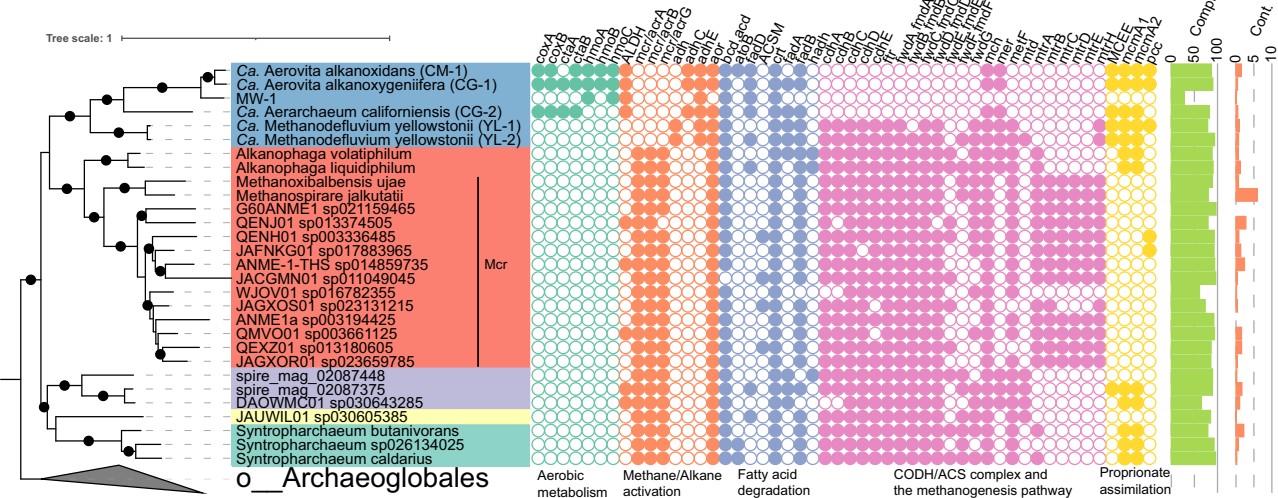

**Fig. 3 | Taxonomy and metabolic comparisons of Syntropharchaeia.** Genome tree and predicted metabolism of representative MAGs for every genus in Syntropharchaeia. Mcr denotes MAGs that encode methane oxidising Methyl-Coenzyme A reductase complex. Filled circles indicate that the gene is encoded by the genome. Compl. and Cont. denote completeness and contamination percentages based on CheckM2 assessment. See Supplementary Data 3 for detailed annotation information.

genes essential for bacterial aerobic methane oxidation, including methanol dehydrogenase (*mdh*), methylene tetrahydrofolate dehydrogenase (*folD*), and methylene-tetrahydromethanopterin dehydrogenase (*mtdB*), were absent in g_CM-1, suggesting methane is unlikely to be its primary substrate. No genes encoding glucose-methanol-choline oxidoreductase, typically required for processing acetol, the product of acetone oxidation were identified. Similarly, we did not detect hydroxylamine dehydrogenase (*hao*) genes characteristic of ammonia-oxidizing bacteria[47], indicating bacterial ammonia oxidation is improbable. In ammonia-oxidizing archaeal (AOA) lineages *Ca.* 'Angelarchaeales' and 'Nitrososphaerales', high copy numbers of conserved plastocyanin-like proteins were found and putatively replace *hao*[48], though their function remains to be determined. While plastocyanin-like proteins were identified in g_CM-1 (Supplementary Data 4), they showed higher sequence similarity to non-AOA bacterial and archaeal lineages, suggesting they are not involved in ammonia oxidation.

Although the specific hydrocarbon(s) used by g_CM-1 HMO complex remains unknown, metabolic inferences suggest that the alcohol byproducts of hydrocarbon oxidation can be fed into the tricarboxylic acid (TCA) cycle (Fig. 4 and Supplementary Fig. 4). All members of f_CG-2 encode genes for aldehyde/alcohol dehydrogenases (ADH/AdhC/AdhE), aldehyde dehydrogenase (ALDH), as well as complete propionate to succinate and beta-oxidation pathways (Figs. 3 and 4). This indicates that f_CG-2 members can metabolise various alcohols. In addition, the presence of long-chain fatty acyl-CoA synthetase (FadD) allows long-chain fatty acids to be used as carbon and energy sources.

Members of f_CG-2 encode a complete electron transport chain, including a truncated complex I (NUO) (Supplementary Fig. 5) that lacks key gene subunits to interact with NADH (*nuoEFH*)[49] or $F_{420}H_2$ (*fpoF*)[50]. Therefore, this complex likely reduces the menaquinone pool and mediates ferredoxin oxidation[51]. Annotated genes encoding complex II (succinate dehydrogenase; SdhABCD) may also mediate the reduction of the menaquinone pool with the conversion of succinate to fumarate. In g_CM-1 MAGs, genes encoding the Etf-like protein complex (FixABCX) likely oxidise NADH and bifurcate electrons for the reduction of ferredoxin and the menaquinone pool[52] (Fig. 4). Genes encoding [2Fe-2S] ferredoxin, known to be more oxygen-tolerant[53] were identified exclusively in members of f_CG-2. Although the specific redox partner for this ferredoxin could not be determined, its

absence in other members of Syntropharchaeia suggests a potential role in aerobic respiration, and an electron donor source for the HMO complex. Oxidation of the reduced menaquinone pool may be mediated by complex III (cytochrome $bc_1$ complex) with the electrons transferred via a cytochrome C carrier protein to complex IV (Cox-ABCD), which catalyses the reduction of oxygen, coupled with proton translocation. The resulting proton gradient drives the ATP production through an archaeal-type ATP synthase (complex V).

## Anaerobic mixotrophy in f_YL-1

In contrast to f_CG-2, the f_YL-1 MAGs encoded genes for the Wood-Ljungdahl pathway (WLP) but lacked the tetrahydromethanopterin S-methyltransferase (Mtr) and MCR/ACR complexes (Figs. 3 and 4). Notably, the *mtrH* gene which is crucial for transfer of methyl groups[54] was identified, linking methyl-CoM and the methyl-branch of the Wood-Ljungdahl pathway. Thus, members of f_YL-1 are unlikely to oxidise methane or hydrocarbons, instead using the Wood-Ljungdahl pathway to fix carbon dioxide. The presence of the WLP genes alongside group 3b and group 4g [NiFe] hydrogenases (Fig. 4) suggests the capacity for lithoautotrophic growth using $H_2$ as an electron donor. The group 3b hydrogenase mediates the reversible oxidation of $H_2$ to the reduction of NADPH, and potentially to the reduction of an unknown electron carrier, which has not been characterised in vivo[55]. In both the YL-1 and YL-2 MAGs, the group 3b gene clusters were co-localised with a heterodisulfide reductase A (*hdrA*) subunit (Supplementary Data 4). Electrons from the group 3b hydrogenase could be transferred to HdrA which is involved in flavin-based electron bifurcation, mediating the reduction of ferredoxin and sending electrons to hdrBC, which mediates the reduction of CoB-S-S-CoM[56]. The group 4g hydrogenase, though not yet biochemically characterised[57], encodes an antiporter-like membrane subunit that suggests sodium/proton translocation. In f_YL-1, the group 4g hydrogenase gene clusters were found adjacent to a coenzyme $F_{420}$-reducing [NiFe]-hydrogenase B subunit (Supplementary Data 4), which mediates $F_{420}$ reduction[58], suggesting $H_2$ oxidation could be coupled to the reduction of $F_{420}$ in contrast to ferredoxin seen typically in other group 4 NiFe hydrogenases in methanogens[59]. Additionally, soluble $F_{420}H_2$ dehydrogenase subunit F genes were identified, potentially enabling reversible electron transfer between $F_{420}H_2$ and ferredoxin. Like f_CG-2, f_YL-1 also possess genetic potential to metabolize long-chain fatty acids, propionate, and alcohols, indicating the capacity for

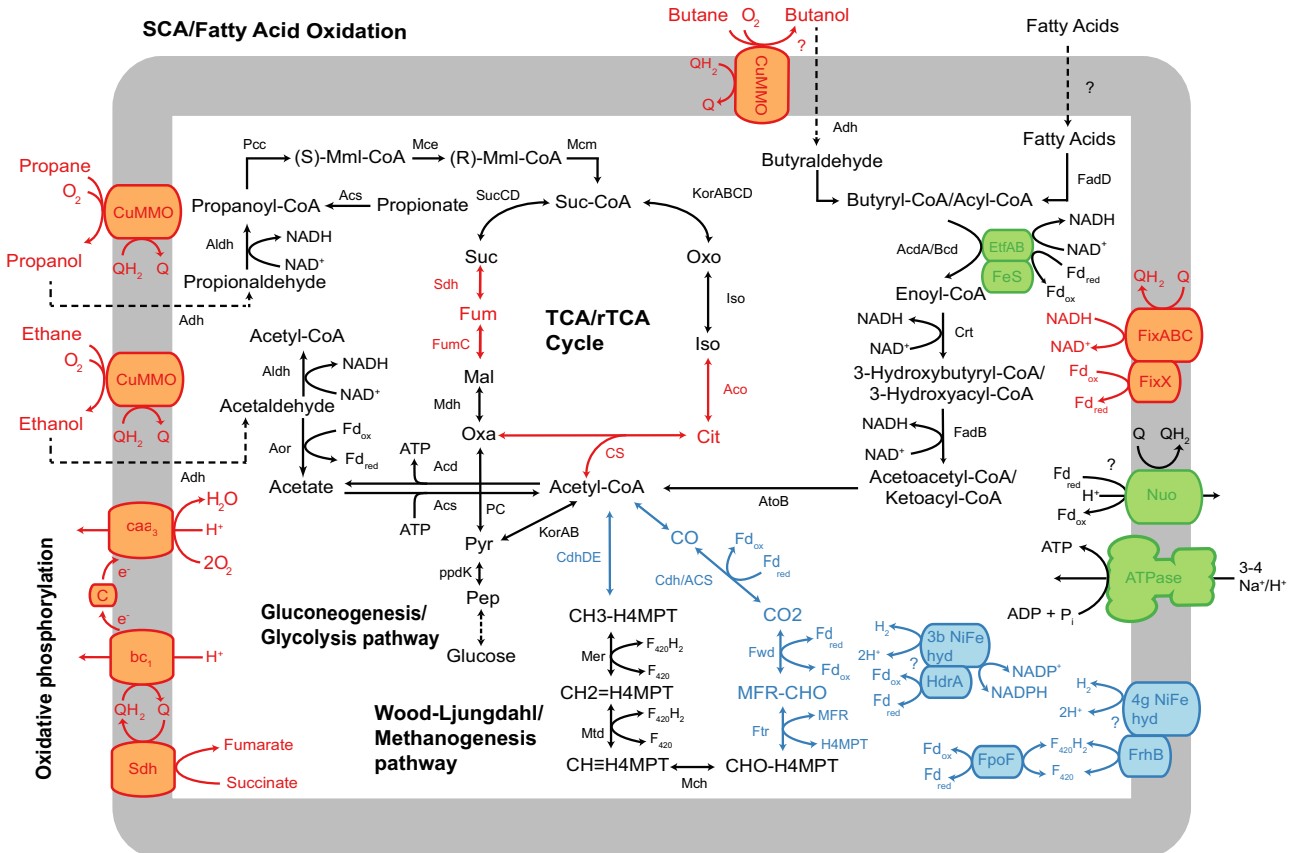

**Fig. 4 | Predicted metabolic capacity of Ca. 'Aerarchaeales'.** Metabolic reconstruction of aerobic and anaerobic members within Ca. 'Aerarchaeales'. Representative genomes CG-1 and YL-1 were used as reference for the metabolic cartoon to represent the *Ca.* 'Aerovita' and *Ca.* 'Methanodefluvium' lineage, respectively. Genes related to hydrocarbon and beta oxidation, the TCA/rTCA cycle, the oxidative phosphorylation chain, the Wood-Ljungdahl pathway, the methanogenesis pathway, and the electron transport chain are shown. Black or green arrows and text represent genes that were identified in both CG-1 and YL-1. Red or Blue arrows and text represent genes that were identified exclusively in CG-1 or YL-1, respectively. CuMMO particulate hydrocarbon monooxygenase, Sdh Succinate

dehydrogenase Complex II, Fpo F420-dehydrogenase Complex I, bc1 Cytochrome bc1 complex, aa3 Cytochrome aa3 complex, Fix electron transfer flavoprotein fix complex, Etf electron transfer flavoprotein complex, Fd ferredoxin, F420 8-hydroxy-5-deazaflavin, NAD+ Nicotinamide adenine dinucleotide, NADP+ Nicotinamide adenine dinucleotide phosphate, NpdG NADPH-dependent F420 reductase, C c-type cytochrome, FpoF $F_{420}H_2$ dehydrogenase subunit F, FrhB coenzyme $F_{420}$-reducing [NiFe]-hydrogenase B subunit, HdrA heterodisulfide reductase subunit A, NiFe hyd [NiFe] hydrogenase. For full names and gene identifications in the CG-1 and YL-1 MAG, see Supplementary Data 4.

mixotrophic growth on both inorganic and organic carbon sources. In contrast to f_CG-2, the f_YL-1 MAGs do not encode a complete electron transport chain or TCA cycle. Instead, they likely use an incomplete rTCA cycle (Supplementary Fig. 5) to funnel acetyl-CoA from the Wood-Ljungdahl pathway and other heterotrophic pathways into the other universal precursors of anabolism (e.g., pyruvate, phosphoenolpyruvate, oxaloacetate, and 2-oxoglutarate)[60].

### Naming of the candidate novel lineages within Syntropharchaeia

Based on the presence of aerobic respiration related enzymes and their phylogenetic placement within the Archaea, we propose the genus *Candidatus* 'Aerarchaeum' (see Supplementary Note 1 for full taxonomy and nomenclature), with *Ca.* 'Aerarchaeum californiensis' for CG-2 as the first representative MAG recovered. The presence of the HMO complex in g_CM-1 supports their putative ability to aerobically oxidise hydrocarbons, and we propose the name *Ca.* 'Aerovita alkanoxidans' for CM-1 as the first representative of a new genera within *Ca.* 'Aerarchaeaceae', and *Ca.* 'Aerovita alkanoxygeniifera' for CG-1, a distinct species within *Ca.* 'Aerovita'. We propose the name *Ca.* 'Methanodefluvium yellowstonii' for YL-1, as the representative strain of a new family that is undergoing loss of the methanogenesis pathway and was recovered from Yellowstone Lake sediments.

### Evolution of metabolism in Syntropharchaeia

To explore the evolutionary processes that led to aerobic hydrocarbon oxidation in *Ca.* 'Aerovita', we applied probabilistic ancestral gene content reconstruction to infer the number of duplication, transfer, and loss events within Syntropharchaeia. Genes encoding the oxygen-sensitive aldehyde:ferredoxin oxidoreductase (AOR), beta-oxidation, and the Wood-Ljungdahl pathway were predicted to be encoded by the last common ancestor (LCA) of Syntropharchaeia (Fig. 5). This supports aldehydes and fatty acid metabolism and the ability to fix $CO_2$ as early metabolic traits within this lineage. Genes encoding the methanogenesis pathway, except for the MCR complex, were inferred to be present in LCA of Syntropharchaeia (Fig. 5; Supplementary Data 5). The loss of many of these genes in extant species suggests that members of Syntropharchaeia are in the process of losing the methanogenesis pathway, which has been independently documented in other archaeal lineages[61]. HGT has subsequently allowed the use of components of the methanogenesis pathway for new metabolic processes. For example, the Syntropharchaeia family *Methanospiraceae* (formerly ANME-1) is hypothesised to have acquired the MCR complex (Fig. 5; Supplementary Fig. 6), allowing them to become anaerobic methane oxidisers. This acquisition is speculated to be via HGT from a relative of *Ca.* Nuwarchaeales[62]. The ACR complex is hypothesised to be present in the LCA of the DAOWMC01, JAUWIL01, and

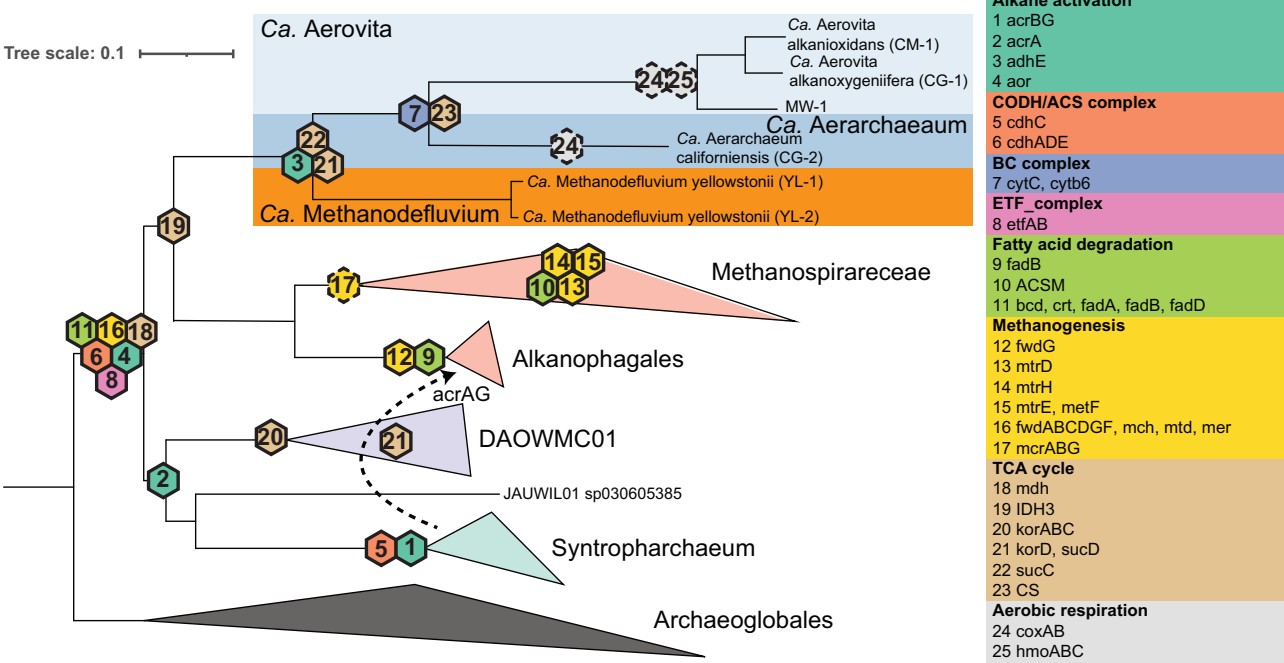

**Fig. 5 | Evolution of Syntropharchaeia metabolism.** Acquisition events of key metabolisms detected through reconciliation of the archaeal species tree with individual gene trees. Acquisition events are denoted with solid hexagons. Dashed hexagons denote inferred acquisition events based only on gene tree analysis. Dashed arrows denote HGT events. Marker genes of the selected metabolisms are supplied in parenthesis. *acrABG* (alkyl-coenzyme M reductase), *adhE* (acetaldehyde dehydrogenase/alcohol dehydrogenase), *aor* (aldehyde ferredoxin oxidoreductase), *cdhADCE* (CODH/ACS complex), *cytC* (C-type Cytochrome), *cytb6* (Cytochrome b6f), *etfAB* (Electron transfer protein), *fadB* (enoyl-CoA hydratase), *ACSM* (medium-chain acyl-CoA ligase/lipoate-activating enzyme), *bcd* (butyryl-CoA dehydrogenase), *crt* (crotonase), *fadA* (acetyl-CoA acyltransferase), *fadB* (enoyl-CoA hydratase), *fadD* (long-chain acyl-CoA synthetase), *fwdABCDGF* (formylmethanofuran dehydrogenase), *mtrDHE* (tetrahydromethanopterin S-methyltransferase), *metF* (methylenetetrahydrofolate reductase), *mch* (methenyltetrahydromethanopterin cyclohydrolase), *mtd* (methylene-tetrahydromethanopterin dehydrogenase), *mer* (5,10-methylenetetrahydromethanopterin reductase), *mdh* (malate dehydrogenase), *IDH3* (isocitrate dehydrogenase), *korABCD* (2-oxoglutarate ferredoxin oxidoreductase), *sucCD* (succinyl-CoA synthetase), *CS* (citrate synthase), *coxABC* (haem-copper oxygen reductase), *hmoABC* (hydrocarbon monooxygenase), *mcrABG* (methyl-coenzyme M reductase). Light blue denotes *Ca.* 'Aerovita' MAGs, blue denote *Ca.* 'Aerarchaeaum californiensis', and orange denote *Ca.* 'Methanodefluvium Yellowstonii'. Blue and orange shading denote aerobicity and anaerobicity based on the presence or absence of the *coxAB* genes.

Syntropharchaeum (Fig. 5), allowing anaerobic hydrocarbon oxidation. The ACR complex was later transferred from Syntropharchaeum to the LCA of the Alkanophagales (Fig. 5; Supplementary Data 5). Together, these results suggest members of the Syntropharchaeia have undergone multiple gain and loss of genes involved in hydrocarbon metabolism.

In contrast to other orders in Syntropharchaeia, the *Ca.* 'Aerarchaeales' have not co-opted the methanogenesis pathway for anaerobic hydrocarbon oxidation. Instead, genes encoding aldehyde-alcohol dehydrogenases (AdhE) are predicted to have been found in the LCA of this lineage (Fig. 5), allowing the use of alcohols and aldehydes as additional substrates. Members of the family *Ca.* 'Aerarchaeaceae' have acquired a complete tricarboxylic acid (TCA) cycle and oxidative phosphorylation pathway, which includes a haem-copper oxygen reductase, supporting their transitions towards an aerobic metabolism (Fig. 5). Gene synteny of the haem-copper oxygen reductase operon and phylogenetic analysis of the CoxA and CoxB subunits suggests two independent acquisition events involving complex IV within *Ca.* 'Aerarchaeaceae' (Supplementary Figs. 7 and 8). Phylogenetic analysis of heme A synthase (*ctaA*) (Supplementary Fig. 9) also supports two acquisition events. However, only a singular gene transfer event is seen in heme O synthase (*ctaB*) for *Ca.* 'Aerarchaeaceae' (Supplementary Fig. 10). The evolutionary processes that led to aerobic metabolism in *Ca.* 'Aerarchaeaceae' remains unclear, with more genomic representatives required to better understand this transition.

Finally, the acquisition of the HMO complex in members of *Ca.* 'Aerovita' suggests that aerobic hydrocarbon oxidation is a relatively recent adaptation. Aerobic oxidation of hydrocarbons in *Ca.* 'Aerovita' enables the production of alcohol, which can then be metabolised through alcohol metabolism pathways that are found in all *Ca.* 'Aerarchaeales'.

### Distribution of Ca. 'Aerarchaeales' in terrestrial and marine environments

To investigate the distribution of *Ca.* 'Aerarchaeales', we searched for closely related 16S rRNA gene sequences (≥90% identity) using the SILVA[63] 138.1 release, revealing five sequences from terrestrial and marine ecosystems (Fig. 6). However, a broader search using the Integrated Microbial Next Generation Sequencing Database (IMNGS)[64] identified members of *Ca.* 'Aerarchaeaceae' and *Ca.* 'Methanodefluviaceae' in 247 samples across a wide range of environments, including aquatic habitats, freshwater sediments, groundwater, soil, and marine ecosystems (Fig. 6). Notably, *Ca.* 'Aerarchaeaceae' was present in 138 samples, while *Ca.* 'Methanodefluviaceae' appeared in 97 samples, with the two lineages only cooccurring in 12 environments, suggesting niche differentiation, presumably due to varying oxygen availability.

Expanding the search to ~250,000 public metagenomes using single copy marker genes from the recovered *Ca.* 'Aerarchaeales' genomes, we identified *Ca.* 'Aerarchaeales' in 64 datasets from similar environments to those identified from the IMNGS survey (Fig. 6). *Ca.* 'Aerarchaeaceae' and *Ca.* Methanodefluviaceae lineages were not

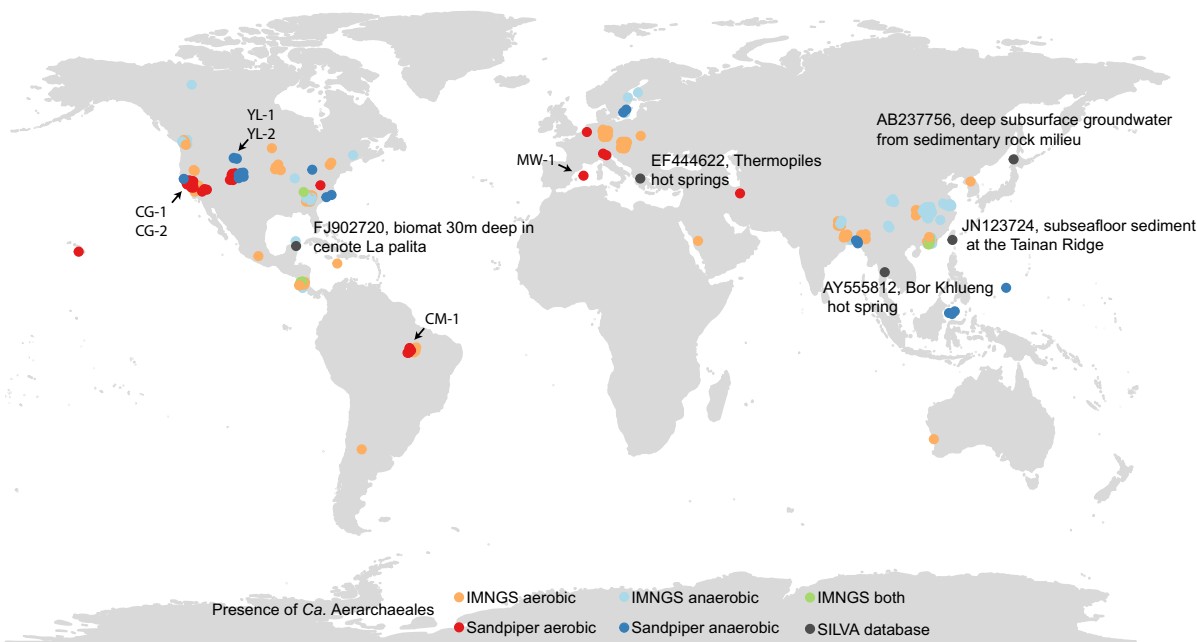

**Fig. 6 | Global occurrence of the *Ca*. Aerarchaeaceae and *Ca*. Methanodefluviaceae lineage.** 16S rRNA genes sequences of the MAGs were searched against the IMNGS database. Sequences (>200 bp) with similarity higher than 95% were regarded as positive hits and the co-ordinates of the environment were depicted in light colours. Co-ordinates of closely related 16S rRNA gene sequences from the SILVA 138.1 data (see Fig. 1B) were depicted in black and the environments labelled. Metagenomes where the *Ca*. 'Aerarchaeales' MAGs were recovered from are noted. Presence and absence of *Ca*. 'Aerarchaeaceae' and *Ca*. 'Methanodefluviaceae' lineages based on Sandpiper were depicted in dark colours. Red, blue, and green dots denote the presence of *Ca*. 'Aerarchaeaceae' and *Ca*. Methanodefluviium, or both lineages. See Supplementary Data 6 and 7 for abundance and additional metadata information.

found together in any of these metagenomes, supporting niche differentiation.

Despite their presence in diverse ecosystems, the relative abundance of *Ca*. 'Aerarchaeales' was consistently low, accounting for <0.1% of the microbial community (Supplementary Data 6 and 7). Further sequencing efforts will be required to elucidate the full metabolic diversity and evolutionary history of this lineage, as well as understand their distribution and importance to global biogeochemical cycling.

## Discussion

*Ca*. 'Aerovita' is the first archaeal lineage predicted to be capable of aerobic hydrocarbon oxidation, a process that was previously assumed to be exclusive to Bacteria. The sequential acquisition of alcohol dehydrogenases, a complete electron transport chain including oxygen reductase, and HMO complex, highlights the impact HGT has had on the metabolic evolution of this lineage.

The acquisition of MCR and ACR complexes within the Syntropharchaeia underscores the dynamic evolutionary history of anaerobic hydrocarbon metabolism in Archaea[19]. Our phylogenetic analysis identified multiple divergent bacteria phyla that encode HMO complexes, suggesting inter-phyla and inter-domain gene transfer of this complex has likely occurred multiple times.

This observation raises important questions about the mechanisms of CuMMO activation across diverse lineages. In aerobic methanotrophic and alkanotrophic bacteria, pMMO/HMO activity is stimulated by reduced quinones[65], which are generated via NADH oxidation and quinone reduction by complex I[66]. In contrast, Nitrososphaerales AOA and *Ca*. 'Angelarchaeales' within Thermoproteota and Thermoplasmatota, respectively are thought to generate reducing equivalents for AMO via alternative mechanisms. These include hydroxylamine oxidation coupled with quinone reduction by blue copper proteins[48,67], or ferredoxin oxidation via a truncated complex I

that lacks the NADH-binding domain[68]. In *Ca*. 'Angelarchaeales', additional ferredoxin-oxidizing systems have been observed, such as a FixABCX complex that couples the NADH oxidation to the simultaneous reduction of ferredoxin and quinone[48]. In *Ca*. 'Aerovita', we identified a truncated complex I, a FixABCX complex, and an oxygen-tolerant [2Fe-2S] ferredoxin, suggesting a distinctive ferredoxin-dependent metabolic configuration for CuMMO activation in archaea. These findings point to an underappreciated diversity in archaeal respiratory strategies, likely shaped by repeated HGT events.

Given the limited number of MAGs in *Ca*. 'Aerarchaeaceae', it is likely that we are underestimating the metabolic repertoire of this lineage. However, even with our current understanding, their detection across diverse marine and terrestrial environments suggests these archaea may contribute to carbon cycling processes in various oxic zones. Future work aimed at increasing genomic representation and cultivation of members of this lineage will be important in confirming their metabolism, in situ activity, as well as developing a better understanding of their ecological importance.

## Methods

### Metagenome assembly and genome binning

For the CM-1 MAG, paired-end read sets (listed in Supplementary Data 1) were downloaded from the NCBI sequence read archive, trimmed and quality filtered using SeqPurge v2018_04[69]. For metagenomic assembly, the paired-end read sets were coassembled using Metaspades, version 3.10.0 using the default parameters[70]. Mapping of quality reads were performed using BamM v1.7.3 with default parameters (https://github.com/Ecogenomics/BamM). Metagenomic assembled genomes were recovered from the assembled metagenomes using uniteM v0.0.14 (https://github.com/dparks1134/UniteM). The CM-1 MAG was further refined by reassembling the mapped quality trimmed reads using SPAdes v3.10.0 using the −careful and

−trusted-contigs setting. The CM-1 MAG was subsequently analysed using RefineM v.0.0.24[71] to identify contigs with divergent tetra-nucleotide frequencies and GC content. For CG-1 and CG-2, all paired-end read sets were coassembled with megahit v1.2.9 and binned using aviary[72] v0.2.0. For MW-1, YL-1, and YL-2, assembly and recovery were performed using aviary v0.2.0 (https://github.com/rhysnewell/aviary) independently for each metagenome. Bins were initially characterised using the classify_wf command with GTDB-Tk[29] v2.4.0. Completeness and contamination rates of the population bins were assessed using CheckM2[27] v1.0.2. Adjusted genome size was calculated as:

$$Adjusted\ genome\ size = \frac{Genome\ size}{Completeness*(1+Contamination)}$$

### Functional annotation

For all MAGs, genes were called and annotated using Prokka v1.14.6[73] using the --kingdom Archaea and --metagenome setting. Additional annotation was performed using the blastp 'verysensitive' setting in Diamond v0.9.30.131[74] against UniRef100 (accessed September 2019)[75], clusters of orthologous groups (COG)[76], Pfam 31[77] and TIGRfam 15.0[78]. Genes were assigned KO ID and key metabolic pathways were identified using KofamScan[79] v1.3.0 using HMM models downloaded 26_04_2024. Genes of interests were further verified using the standalone RPS-BLAST v2.12.0 against the NCBI conserved domain database using default parameters[80] to identify conserved motif(s). Putative hydrogenases were classified using the HMM profiles from HydDB[57] (GitHub - GreeningLab/HydDB: Greening lab hydrogenase database) using default settings. Contigs encoding the HMO-like complex were visualised using the gggenomes[81] package v1.0.1 in R v4.3.3.

### Identification of AOA plastocyanin-like proteins

Plastocyanin-like proteins in AOA were identified following methods similar to Diamond et al.[48]. We detected these proteins by screening for Pfam domains within the Pfam CU_oxidase clan (CL0026), specifically targeting: Copper-bind, COX2, COX_ARM, Cu-oxidase, Cu-oxidase_2, Cu-oxidase_3, Cupredoxin_1, Cu_bind_like, CzcE, DP-EP, Ephrin, hGDE_N, PAD_N, PixA, and SoxE domains. To determine their evolutionary relationships, we compared the identified putative plastocyanin-like proteins against reference genomes from GTDB[82] v220 genomes to identify their closest homologs.

### Construction of genome tree

The archaeal genome tree was constructed using GTDB-Tk[29] v2.4.0 using the de_novo_wf command with p_Undinarchaeota as the outgroup taxon. Briefly, 52 single copy marker genes were identified and aligned in each genome, concatenated, and trees were constructed using FastTree[64] v2.1.11 with the WAG + GAMMA models. Support values were determined using 100 nonparametric bootstraps. The trees were visualised using iTOL[83] v6 and modified using Adobe illustrator. The genome tree shown in Fig. 1A includes only the best representative species for each genus within the Halobacteriota. Taxonomic ranking of MAGs was inferred based on relative evolutionary divergence[82] using GTDB-Tk with the infer_ranks command and average nucleotide identity (ANI%). Pairwise ANI calculations between genomes were calculated using skani[84] v0.1.4.

### Construction of 16S rRNA gene tree

The 16S rRNA gene sequences identified in the genomes using Barrnap v0.9 (https://github.com/tseemann/barrnap) were used to infer taxonomic assignment. Sequences were compared against the SILVA[63] 16S rRNA database (Version 138.1). Sequences were aligned with 291 16S rRNA sequences retrieved from the SILVA database using muscle[85] v5.01278 using the -maxiters 2 parameters. The phylogenetic tree was constructed using IQ-TREE[86] v2.3.3 --ufboot 1000 -nt AUTO -m MFP -T

AUTO. Trees were visualised using iTOL[83] and modified using Adobe illustrator.

### Construction of the CuMMO and other gene trees

The CuMMO tree shown in Fig. 2A was generated from concatenated A, B, and C subunit genes for each operon. Genes were sourced from reference sequences that cover the known diversity of these protein subunits[48,87]. Top homology hits against our HMO-like sequences from the GTDB[82] v220 genomes were also added. Genes were considered in an operon if their gene IDs were in sequential order, and incomplete operons were excluded. Each individual protein subunits were aligned using muscle[85] v5.01278. The final concatenated CuMMO subunits comprised of 129 sequences. The phylogenetic tree was inferred using pargenes[88] v1.2.0 under default settings. Briefly, ModelTest-NG[89] v0.1.7 was used to select the best-fit model for the protein alignments, and the maximum-likelihood tree was generated using RAxML-NG[90] v1.0.1. Support values were determined using 100 nonparametric bootstraps. Individual phylogenetic gene trees were generated for *mcrA*, *coxA*, *coxB*, *ctaA*, and *ctaB* in the same way.

### Structural characterisation of the CuMMO complex

The protein sequence of the CuMMO A, B, and C subunits were used for structural characterisation using Colabfold[35] local v1.5.2 in multimer mode. The MW-1 CuMMO complex is truncated (likely due to MAG incompleteness) and hence was not included in this analysis. 3D protein structures figures were generated using Mol* Viewer[91] v4.17.0 and pyMOL[92] v2.5.7. pLDDT colour schemes were generated using the custom script (https://github.com/ailienamaggiolo/alphafold_coloring). Figures were further refined using Adobe Illustrator.

### Structural annotation

Predicted structures in pdb format were searched using Foldseek[93] v8-ef4e960 multimer to find structural homologies against the pdb100 and CATH50 database. Domains were considered significant if they had a probability of $p \geq 0.9$ and an e-value less than 0.01. Alphafill[94] v2.1.0 was used to enrich our predicted 3D structures with ligands and cofactors. In the CM-1 and CG-1 HMO complexes, two mononuclear copper ions were confidently placed in the CuB and CuC sites with high confidence local r.m.s.d. and transplant clash scores (Supplementary Data 2).

### Gene tree-aware ancestral reconstruction

Protein families of interest, annotated with KO IDs, were used for ancestral reconstruction analysis. Only gene families with ≥4 sequences and ≥30 amino acids were used for further analysis. Each protein family was aligned using muscle[85] v5.01278, processed with trimAl[95] v1.4.rev15 with the automated1 setting, and maximum-likelihood phylogenetic trees were constructed for each alignment using IQ-TREE[86] v2.3.3 with parameters "-m MFP -bb 10000 -nm 20000 -T AUTO –wbtl". Multifurcations for each protein family trees were corrected using Treerecs[96] v1.2. The protein trees were probabilistically reconciled against the supermatrix species tree and sampled 100 times with the ALEml_undated function of the ALE[97] package v1.0 to infer numbers of duplications, transfers, losses, and originations on each branch of the supermatrix tree. The species tree used for reconciliation was derived from the genome tree shown in Fig. 1A by removing all genomes except MAGs from the Syntropharchaeia and the Archaeoglobales as the taxon outgroup. Genome completeness assessed by CheckM2[27] was used to reduce estimation bias caused by incomplete genomes. ALE outputs were filtered using a frequency threshold of >0.3 to identify events, to account for potential noise that may arise from sequence alignment and tree reconstructions. The probable origination events were used to generate Fig. 5. Potential horizontally transferred genes were further validated by additional phylogenetic analysis.

### Environmental distribution analysis

The 16S rRNA gene sequences identified in our MAGs were used as queries in the IMNGS[64] database (Version 1.0 Build 2508) to explore the global occurrence of the aerobic and anaerobic lineages, *Ca.* 'Aerarchaeaceae' and *Ca.* 'Methanodefluviaceae', respectively. The similarity threshold was set at 95% ANI and sequence length threshold was set at 200 bp.

To search the Sandpiper database for metagenomes containing our MAGs, SingleM[23] v0.18.0 "supplement" was used to add our genomes to the GTDB R220-derived S4.3.0 metapackage (excepting CG-2 since that is already included in the metapackage), using "--new-fully-defined-taxonomies" to define taxonomy according to Fig. 1, and "--no-quality-filter --no-dereplication --no-taxon-genome-lengths". SingleM "renew" was then run on the "archive OTU tables" used to generate version 0.3.0 of the Sandpiper database. Samples containing lineages of interest were identified by finding non-zero relative abundance in the resulting taxonomic profiles.

GPS coordinates of environments retrieved from the IMNGS and metagenome databases were used as input to make the global occurrence figure (Fig. 6). This figure was generated using the sf[98] v1.0-21 library package. Closely related OTU sequences from the SILVA database (see Fig. 1B) were also included in this analysis.

### Reporting summary

Further information on research design is available in the Nature Portfolio Reporting Summary linked to this article.

### Data availability

The MAGs assembled in this study have been deposited in the NCBI database under the accession numbers SAMN45772796 [https://www.ncbi.nlm.nih.gov/biosample/45772796] to SAMN45772801 [https://www.ncbi.nlm.nih.gov/biosample/45772801] under the Bioproject ID PRJNA1197096 [https://www.ncbi.nlm.nih.gov/bioproject/1197096].

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

## Acknowledgements

We thank Yosephine Gumulya, Nicholas Coleman, Paul N. Evans, Marc T. Morris, Marcelo M. Pedroso, and Gerhard Schenk for their helpful discussions. We thank Dr. Maria Chuvochina for her valuable guidance and expertise in the nomenclature and taxonomic assignment of the novel archaeal metagenome-assembled genomes (MAGs) described in this study. A.O.L., G.W.T., S.J.M, and B.J.W. were supported by Australian Research Council Fellowships (DE250101094, FL230100159, FT190100211, and FT210100521, respectively).

## Author contributions

A.O.L., B.J.W. and G.W.T conceived the project. A.O.L. wrote the manuscript. A.O.L. and B.J.W. performed the bioinformatic analyses. A.O.L., B.J.W., S.J.M. and G.W.T. contributed to the drafting of the manuscript.

## Competing interests

The authors declare no competing interests.
