## [Transparent Peer Review file · Nature Communications]

Potential for aerobic hydrocarbon oxidation in archaea

Corresponding Author: Dr Andy Leu

Version 0:

Reviewer comments:

Reviewer #1

(Remarks to the Author)

Aerobic hydrocarbon degradation has been strictly associated with bacteria, but Leu et al overturn this through metagenomic discovery of an uncultured lineage of archaea that encodes hydrocarbon monooxygenases associated with aerobic hydrocarbon degradation and other oxygen-dependent enzymes. This is highly novel and of broad interest to microbiology and ecology. However, one of the core conclusions of this study—these archaea take an aerobic lifestyle—lacks sufficient evidence and there are clear misinterpretations that preclude this conclusion. The authors must take more caution in interpretation of the genomic annotation data. Another key conclusion regarding the evolution of these aerobic hydrocarbon-degrading archaea also seems to disagree with the presented data. I believe addressing the concerns below may help unravel the interesting physiology of *Aerovita* that may set them apart from aerobic hydrocarbon-degrading bacteria.

While I agree that *Aerovita* can likely utilize oxygen (e.g., for conversion of alkanes to alcohols), the authors cannot conclude that *Ca. Aerovita* lives aerobically based on their data. Utilizing oxygen, respiring oxygen, and an aerobic lifestyle are different biological phenomena. Having terminal oxidases is not sufficient to conclude that an organism lives aerobically—there are plenty of obligate anaerobes that have terminal oxidases. Moreover, of the enzymes that the authors suggest are involved in energy metabolism / electron transfer, three are oxygen sensitive: (i) the electron-confurcating butyryl-CoA dehydrogenase (or electron-bifurcating crotonyl-CoA reductase) is highly oxygen sensitive, (ii) KorABCD is oxygen sensitive, and (iii) FixABCX is typically associated with anaerobic (e.g., syntrophic fatty acid oxidizers) or micro/nano-aerobic (e.g., nitrogen fixation in root nodules) metabolisms. Clearly, these archaea are facultatively anaerobic and/or live under microoxic conditions where oxygen toxicity is lower. This is quite interesting and definitely worth exploring.

In addition, based on the presented data, I do not agree with the authors' claim that a complete electron transfer pathway is present for aerobic respiration. While the authors claim that FixABCX is part of this pathway, it is very likely sensitive to oxygen. The authors ought to identify an alternative enzyme for transferring electrons from NADH to menquinone (aerobic respiration via complex III and IV requires reduced quinones). Do the authors see a type II NADH:quinone oxidoreductase? Likewise, do the authors see aerotolerant 2-oxoglutarate dehydrogenases that the archaea may use to replace KorABCD under aerobic conditions? The absence of such pathways and presence of the above oxygen-sensitive enzymes would suggest that the archaea depends on micro- or nano-aerobic conditions where it can use oxygen for alkane activation and oxygen respiration, but many electrons are funneled into anaerobic pathways. Note that nitrogen-fixing aerobic bacteria live roughly similar lifestyles.

LN181: How are the authors concluding mixotrophy from the metabolic reconstruction? I do not see any data that clearly supports this conclusion.

LN227-235, LN236-237: The authors claim that there were two separate acquisition events for aerobic respiration enzymes in *Ca. Aerarchaeaceae* in LN227-235 and Fig. 5. More specifically, the analyses seem to suggest aerobic respiration was acquired after the divergence of MW-2 in *Ca. Aerovita*. According to Fig. 2, HMO was acquired by the ancestor of *Ca. Aerovita* before MW-2 diverged. I do not understand how the authors conclude that HMO was acquired after aerobic respiration. Should it not be the other way around?

LN265: Again, I do not see a complete electron transport chain. Moreover, there is no discussion of why *Aerovita* has ferredoxin-dependent enzymes like the electron-confurcating butyryl-CoA dehydrogenase (or electron-bifurcating crotonyl-CoA reductase) and KorABCD.

Minor:

LN134: "Structural characterization" is not an appropriate phrase here (the authors do not characterize but rather predict the structure). I would use "structural prediction" or the like.

LN138: What is meant by "significant"? If statistical data is available, please indicate that. Otherwise use another word.

LN230-232: heme A synthase? Heme O synthase?

LN238-241: I cannot understand what this sentence means.

Reviewer #2

(Remarks to the Author)

RE: 'Aerobic hydrocarbon oxidation in a novel archaeal lineage'

This manuscript proposes aerobic hydrocarbon oxidation by Candidatus 'Aerovita' of Syntropharchaeota based on analysis of metagenome-assembled genomes (MAGs). The authors obtained three MAGs of this new genus and identified copper-containing membrane monooxygenase (CuMMO) genes, suggesting a hydrocarbon oxidation trait for this clade. Based on the presence of oxygen-dependent enzymes, the authors propose a shift from anaerobic to aerobic lifestyle via horizontal gene transfer between archaea and bacteria. The novelty lies in the fact that no archaea have yet been reported to oxidize hydrocarbons aerobically, although Syntropharchaeota contains ACR genes involved in anaerobic alkane oxidation (propane, butane, and long-chain alkanes). Together, aerobic alkane oxidation has only been reported in bacteria, and this study proposes to extend this trait to archaea.

Major comments:

The term "hydrocarbon monooxygenase" (HMO) should be more specific. CuMMO typically refers to short-chain hydrocarbon monooxygenase, while long-chain hydrocarbon monooxygenase contains a diiron center.

CuMMO is known to be promiscuous in terms of substrate. Downstream pathways, rather than the enzyme itself, often determine the physiological roles of CuMMO. Phylogenetic evidence only partially supports the enzyme's physiological function. As the authors noted, many phylogenetically divergent phyla contain CuMMO, but their functions cannot be definitively classified as HMOs without further evidence (e.g., gene expression analysis).

Although alcohol and aldehyde dehydrogenase genes were reported in the MAGs, these are present in multiple copies and can be involved in various metabolic pathways of general heterotrophs. In addition, in the manuscript, it is described that fatty acid utilization is LCA trait of Syntropharchaeota, meaning not related to presence of CuMMO. It would be beneficial to determine if these genes are localized in the pseudoperiplasm, as in other CuMMO cases (e.g., AMO, PMO). Based solely on genomic analysis, without additional supporting evidence, it is challenging to claim this novel trait, which is a key aspect of the manuscript.

While present in various environments as any microbe, the abundance of this archaeal lineage appears to be low, and its role as alkane oxidizers in ecosystems should be carefully considered. General information.

Specific comments:

Line 122: If there is evidence of CuMMO involvement in long-chain alkane oxidation, it should be described in detail and discussed. I don't know about that.

Lines 114-116: Consider whether three MAGs provide sufficient evidence for the claim made.

Lines 145-147: Clarify whether the conservation is specific to HMO or applies to all CuMMO. I think it is conserved in all CuMMO.

Lines 178-179: Elaborate on potential electron donors for CO₂ fixation.

Lines 184-187: Clarify the relationship between incomplete electron transport and the presence of rTCA.

Lines 202-204: Beta oxidation pathway enzymes themselves may not be sensitive to oxygen. If beta oxidation is a trait of the last common ancestor of Syntropharchaeota, it is not unique to the CuMMO MAGs, potentially weakening the connection between CuMMO and fatty acid oxidation.

Version 2:

Reviewer comments:

Reviewer #1

(Remarks to the Author)

I appreciate the authors' responses and agree that revising the term "aerobic lifestyle" to "aerobic respiration" improves clarity. I also concur with the conclusion that these archaea are very likely capable of oxygen-dependent hydrocarbon degradation—a novel and intriguing finding—and that they can respire oxygen. However, I still find issues in the reconstruction and interpretation of these archaea's metabolism that need to be addressed.

As noted in my previous comments, the aerobic metabolism described for this archaeal lineage is atypical and warrants

explicit clarification. Without sufficient explanation, the proposed electron transport system appears confusing, especially in comparison to typical aerobic respiration pathways found in model bacteria. According to the authors' data, a key respiratory complex—complex I (NADH dehydrogenase)—lacks important subunits (e.g., NuoEF; see Extended Data Fig. 5). The authors infer that this truncated complex instead oxidizes ferredoxin. Based on their annotations (e.g., Fig. 4), this partial complex I and FixABCX are the only candidate electron carrier-oxidizing enzymes that could supply electrons into the quinone pool for aerobic respiration. This implies that ferredoxin is centrally involved in the oxygen respiratory chain of these archaea—an unusual feature, given ferredoxin's general oxygen sensitivity.

In my previous comments, I noted that this observation is at odds with the aerobic lifestyle the authors attribute to these organisms. The authors disagree, arguing that the presence of ferredoxin-dependent enzymes does not imply facultative anaerobiosis or microaerophily, citing examples of "obligate aerobe" that use ferredoxin (Response 1.3). However, the cited examples are not broadly representative. Specifically:

1. *S. solfataricus* and *H. salinarum* can grow anaerobically via anaerobic respiration (see 10.1099/ijsem.0.002665 and 10.3389/fmicb.2022.811432/full).
2. These archaea inhabit extreme environments (high temperature or high salinity) where oxygen solubility is low—*Sulfolobus* is hyperthermophilic, *Halobacterium* and *N. magadii* are halophilic (see 10.1038/npg.els.0000394).
3. In some cases, ferredoxin is used in conjunction with protective strategies against oxygen, such as cell-coating with alginate, as in *Azotobacter vinelandii*, which also uses ferredoxin for nitrogen fixation (see 10.1128/AEM.66.9.4037-4044.2000). The use of FixABCX in these taxa is also linked to low-oxygen adaptations like nitrogenase function.

In other words, these examples of "obligate aerobe" are not actually paradigms of high-oxygen adaptation, and they share ecological or physiological features that allow the coexistence of oxygen sensitivity and respiration. Critically, none of the environments from which the authors' MAGs are derived appear to share such extreme or microoxic characteristics. This undermines the generalizability of the cited examples and emphasizes the uniqueness of the finding.

Indeed, the study the authors themselves cite (Yan et al., *Sci Rep* 2016) notes that oxygen-sensitive OFORs are "currently only observed in aerobic organisms living in extreme conditions, such as high temperature or high salt concentrations." The present MAGs are neither. Therefore, rather than rebutting the concern, the cited cases actually reinforce that the co-occurrence of ferredoxin and oxygen respiration outside of extreme environments is uncommon—and potentially notable. I strongly recommend that the authors frame this feature as a novel metabolic configuration that deserves attention (e.g., thermophilic, halophilic, and now this lineage).

In Response 1.3, the authors refer to KEGG data showing ferredoxin-linked enzymes in other aerobic archaea. Could the authors further examine and report on the ecological distribution of these archaea to evaluate whether they also occupy low-O₂ or extreme environments? My guess is that most are halophiles or hyperthermophiles. Again, this context would highlight what makes their current archaeal group distinctive.

In addition, a comparison with canonical aerobic hydrocarbon-degrading bacteria would strengthen the manuscript. These bacteria typically employ a very different metabolic setup (e.g., full complex I, NADH as electron donor, etc.). A few sentences contrasting these bacteria with the archaea studied here would help readers appreciate the unusual nature of this archaeal metabolism.

Regarding Cox as a marker for aerobic respiration, the authors state it is associated with respiration at higher oxygen concentrations (Response 1.2). However, recent evidence suggests otherwise—an obligately anaerobic archaeon and other anaerobic bacteria also encode Cox (see 10.1101/2025.02.26.640444), indicating that some Cox types may function under low-O₂ conditions. This weakens its utility as a definitive marker of "aerobic lifestyle."

Similarly, the authors suggest that the absence of the Wood–Ljungdahl pathway supports an aerobic lifestyle. This is not a valid inference; many anaerobes lack this pathway, and its absence does not equate to aerobic metabolism.

Finally, regarding Fig. 4: the depiction of electron-bifurcating butyryl-CoA dehydrogenase appears to have the directionality reversed. NAD⁺ should be reduced and ferredoxin oxidized during beta-oxidation, not the other way around.

Reviewer #2

(Remarks to the Author)

Reviewer Comments for NCOMMS-25-00654B

General Assessment

The revised manuscript is significantly improved. However, inevitably, the novelty of the discovery is diminished by the fact that the functional claims are based solely on genomic analysis of microorganisms that are distantly related to previously known HMO (CuMMO) lineages, and by their low abundance in the environment (despite being widespread). Phylogenetic and similarity analyses of HMO genes may not be particularly valuable for such distantly related CuMMO microorganisms. That said, given the inherent limitations of MAGs, the manuscript's claims are reasonably supported by the genomic data. The overall narrative and logic of the manuscript remain intact and well-written. I agree with the suggestion that future work should focus on increasing genomic representation and cultivating members of this lineage to confirm their metabolism and ecological importance.

Comments on Authors' Responses

Response 2.1 and 2.6

Regarding the statement: "Therefore, we have chosen to use the more general term hydrocarbon monooxygenase to reflect the broad substrate specificity of these enzymes across various hydrocarbons, rather than limiting it to a specific chain length."

Upon reviewing the four cited references, the evidence indicates that CuMMO/HMO enzymes are limited to short-chain (C2–C4) gaseous hydrocarbons. Please review these references again, as they do not support activity beyond C4 alkanes.

Response 2.2

While the process of elimination suggests a close relationship to HMO microorganisms, with a low likelihood of AMO or PMO involvement, other functional possibilities remain. Many CuMMO enzymes, such as pxmABC, have unknown functions and are widespread, as discussed in the literature. For example, pxmABC is prevalent but its function remains unclear, and other CuMMOs in Alpha-, Beta-, and Gamma-proteobacteria are found near the base of Fig. 1 (the first ref). Recent studies have also suggested acetone monooxygenase activity from PMO-like gene clusters (the second one).

- <https://www.frontiersin.org/journals/microbiology/articles/10.3389/fmicb.2018.02493/full>
- <https://academic.oup.com/isme/article/15/12/3636/7474400>

Response 2.3

I apologize for the earlier lack of clarity. I intended to highlight that alcohol dehydrogenase (ADH) is usually localized outside the cell membrane. Please check the localization of ADH, not CuMMO. If all ADHs are cytoplasmic, you can propose the presence of transporters for alcohol intermediates.

Response 2.9

My previous comment referred not to the direct electron donor, but to the ultimate source of energy (e.g., phototrophy or lithotrophy) for CO₂ fixation. Please clarify whether CO₂ fixation is linked to autotrophic growth in these organisms.

Additional Comments

Line 67 and elsewhere: The term "mixotrophic" is unclear. Please define it at first use.

Figure 1: Please add the order label 'JACQPP01' for clarity.

Extended Figure 2: The figure is almost illegible due to the small font size. Please increase the font size for readability. The (deep) branching of the HMO should also be more clearly visualized.

Line 257: Please clarify whether the ANI% refers to the 16S rRNA gene sequence.

Response to the Reviewers:

Reviewer #1 (Remarks to the Author):

Aerobic hydrocarbon degradation has been strictly associated with bacteria, but Leu et al overturn this through metagenomic discovery an uncultured lineage of archaea that encodes hydrocarbon monooxygenases associated with aerobic hydrocarbon degradation and other oxygen-dependent enzymes. This is highly novel and of broad interest to microbiology and ecology. However, one of the core conclusions of this study—these archaea take an aerobic lifestyle—lacks sufficient evidence and there are clear misinterpretations that preclude this conclusion. The authors must take more caution in interpretation of the genomic annotation data. Another key conclusion regarding the evolution of these aerobic hydrocarbon-degrading archaea also seems to disagree with the presented data. I believe addressing the concerns below may help unravel the interesting physiology or Aerovita that may set them apart from aerobic hydrocarbon degrading bacteria.

R1. Thank you for your thoughtful feedback and for recognising the novelty and significance of our study. We appreciate your insights and the opportunity to clarify our interpretations. However, we respectfully disagree with some of your points and hope to provide compelling evidence that our metabolic reconstruction has been carefully conducted and appropriately interpreted.

While I agree that Aerovita can likely utilize oxygen (e.g., for conversion of alkanes to alcohols), the authors cannot conclude that *Ca. Aerovita* lives aerobically based on their data. Utilizing oxygen, respiring oxygen, and an aerobic lifestyle are different biological phenomena. Having terminal oxidases is not sufficient to conclude that an organism lives aerobically—there are plenty of obligate anaerobes that have terminal oxidases.

R2. While we acknowledge your distinction between utilizing oxygen, respiring oxygen, and an aerobic lifestyle, we believe that the presence of specific terminal oxidases provides strong evidence that *Ca. Aerovita* is adapted to an aerobic environment.

The presence of terminal oxidases in some obligate anaerobes is indeed well-documented, but these are typically cytochrome bd oxidases, which have a high affinity for oxygen and function under microaerophilic conditions. In contrast, our MAGs encode Cox heme-copper oxidases, which are characteristic of organisms that thrive in higher oxygen concentrations. This suggests that *Ca. Aerovita* is not primarily a microaerophile but rather capable of tolerating and utilizing oxygen at higher levels.

To support this interpretation, we reference recent studies that use the presence of Cox heme-copper oxidases and cytochrome bd oxidases as key indicators of aerobic and microaerophilic lifestyles in novel lineages:

- Sheridan, P. O., et al. (2022). *Recovery of Lutacidiplasmatales archaeal order genomes suggests convergent evolution in Thermoplasmatota*. *Nature Communications*, 13(1), 4110.
- Luo, Z.-H., et al. (2024). *Temperature, pH, and oxygen availability contributed to the functional differentiation of ancient Nitrososphaeria*. *ISME Journal*, 18(1), wrad031.
- Speth, D. R., et al. (2024). *Genetic potential for aerobic respiration and denitrification in globally distributed respiratory endosymbionts*. *Nature Communications*, 15(1), 9682.

Moreover, of the enzymes that the authors suggest are involved in energy metabolism / electron transfer, three are oxygen sensitive: (i) the electron-confurcating butyryl-CoA dehydrogenase (or electron-bifurcating crotonyl-CoA reductase) is highly oxygen sensitive, (ii) KorABCD is oxygen sensitive, and (iii) FixABCX is typically associated with anaerobic (e.g., syntrophic fatty acid oxidizers) or micro/nano-aerobic (e.g., nitrogen fixation in root nodules) metabolisms. Clearly,

these archaea are facultatively anaerobic and/or live under microoxic conditions where oxygen toxicity is lower. This is quite interesting and definitely worth exploring.

In addition, based on the presented data, I do not agree with the authors' claim that a complete electron transfer pathway is present for aerobic respiration. While the authors claim that FixABCX is part of this pathway, it is very likely sensitive to oxygen. The authors ought to identify an alternative enzyme for transferring electrons from NADH to menaquinone (aerobic respiration via complex III and IV requires reduced quinones). Do the authors see a type II NADH:quinone oxidoreductase?

Likewise, do the authors see aerotolerant 2-oxoglutarate dehydrogenases that the archaea may use to replace KorABCD under aerobic conditions? The absence of such pathways and presence of the above oxygen-sensitive enzymes would suggest that the archaeon depends on micro- or nano-aerobic conditions where it can use oxygen for alkane activation and oxygen respiration, but many electrons are funneled into anaerobic pathways. Note that nitrogen-fixing aerobic bacteria live roughly similar lifestyles.

R3. We appreciate the reviewer's detailed feedback and the opportunity to clarify the role of these enzymes in *Ca. Aerovita*. While we acknowledge that certain enzymes involved in energy metabolism and electron transfer can be oxygen-sensitive, we respectfully disagree with the assertion that their presence necessarily indicates a facultatively anaerobic or microaerophilic lifestyle. Below, we provide evidence demonstrating that these enzymes are not strictly anaerobic and have been identified in aerobic archaea and bacteria.

1. Electron-confurcating butyryl-CoA dehydrogenase / electron-bifurcating crotonyl-CoA reductase:

While it is true that this enzyme can be oxygen-sensitive, its presence is not exclusive to anaerobic organisms. This protein has been identified in multiple strictly aerobic archaea across diverse phyla, including *Haloarchaeum salinarum*, *Sulfolobus solfataricus* P2, and *Natrialba magadii*, among others. This suggests that oxygen sensitivity alone does not preclude its function in an aerobic environment.

Reference:

- Dibrova, D.V., Galperin, M.Y., & Mulkidjanian, A.Y. (2014). *Phylogenomic reconstruction of archaeal fatty acid metabolism*. *Environmental Microbiology*, 16(4), 907–918.

2. 2-oxoacid-ferredoxin oxidoreductase (KorABCD):

While some KorABCD are known to be oxygen-sensitive, it has been identified in multiple aerobic archaea and bacteria, suggesting that its presence does not necessarily indicate strict anaerobic metabolism.

References:

- Yan, Z., et al. (2016). *Crystal structures of archaeal 2-oxoacid: ferredoxin oxidoreductases from Sulfolobus tokodaii*. *Scientific Reports*, 6(1), 33061.
- Yoon, K.-S., et al. (1996). *Purification and characterization of 2-oxoglutarate: ferredoxin oxidoreductase from a thermophilic, obligately chemolithoautotrophic bacterium, Hydrogenobacter thermophilus TK-6*. *Journal of Bacteriology*, 178(11), 3365–3368.
- Kerscher, L., & Oesterhelt, D. (1981). *The catalytic mechanism of 2-oxoacid: ferredoxin oxidoreductases from Halobacterium halobium: one-electron transfer at two distinct steps of the catalytic cycle*. *European Journal of Biochemistry*, 116(3), 595–600.

3. FixABCX complex:

While FixABCX is often associated with anaerobic or microaerobic processes, its

presence in *Azotobacter vinelandii*, an obligate aerobe, suggests that it can also function in oxygen-rich environments.

Reference:

- Ledbetter, R.N., et al. (2017). *The electron-bifurcating FixABCX protein complex from Azotobacter vinelandii: generation of low-potential reducing equivalents for nitrogenase catalysis*. *Biochemistry*, 56(32), 4177–4190.

Furthermore, a search of the KEGG genes database (<https://www.genome.jp/kegg/genes.html>) confirms that these enzymes are widely distributed across aerobic archaea, reinforcing the findings from the literature. Given this evidence, we believe it is premature to classify these enzymes as strictly anaerobic or to conclude that *Ca. Aerovita* is incapable of utilizing them in an aerobic capacity.

LN181: How are the authors concluding mixotrophy from the metabolic reconstruction? I do not see any data that clearly supports this conclusion.

The lineage f__YL-1 possesses genes for both the Wood-Ljungdahl pathway (an autotrophic process) and fatty acid metabolism (a heterotrophic process), demonstrating its mixotrophic lifestyle. Since this information is already conveyed in the paragraph, no further clarification is necessary.

LN227-235, LN236-237: The authors claim that there were two separate acquisition events for aerobic respiration enzymes in *Ca. 'Aerarchaeaceae'* in LN227-235 and Fig. 5. More specifically, the analyses seem to suggest aerobic respiration was acquired after the divergence of MW-2 in *Ca. 'Aerovita'*. According to Fig. 2, HMO was acquired by the ancestor of *Ca. 'Aerovita'* before MW-2 diverged. I do not understand how the authors conclude that HMO was acquired after aerobic respiration. Should it not be the other way around?

R4. We appreciate the reviewer's careful evaluation of our conclusions. Based on the context of the manuscript, we assume the reviewer is referring to MW-1 rather than MW-2, as MW-2 is not discussed in the text.

As shown in Figure 5, both the HMO and CoxAB genes are predicted to have been acquired in the last common ancestor of *Ca. Aerovita*. Given this, the precise order of these lateral acquisitions remains uncertain. To address this ambiguity, we have revised the sentence for clarity:

"Finally, the acquisition of the HMO complex in members of Ca. Aerovita suggests that aerobic hydrocarbon oxidation is a relatively recent adaptation."

LN265: Again, I do not see a complete electron transport chain. Moreover, there is no discussion of why *Aerovita* has ferredoxin-dependent enzymes like the electron-confurcating butyryl-CoA dehydrogenase (or electron-bifurcating crotonyl-CoA reductase) and KorABCD.

R5. Please refer to our response in R3, where we have addressed the concerns regarding the electron transport chain and the presence of ferredoxin-dependent enzymes.

Minor:

LN134: "Structural characterization" is not an appropriate phrase here (the authors do not characterize but rather predict the structure). I would use "structural prediction" or the like.

R6. We appreciate the reviewer's suggestion. The phrase has been revised to "structural prediction" to more accurately reflect our methodology.

LN138: What is meant by “significant”? If statistical data is available, please indicate that. Otherwise use another word.

**R7. To provide clarity, we have now included statistical values from Foldseek:
Probability ≥ 0.99 , qTM = 0.9, tTM = 0.31.**

LN230-232: heme A synthase? Heme O synthase?

R8. We have corrected this to "heme A synthase and heme O synthase.

LN238-241: I cannot understand what this sentence means.

**R9. We appreciate the reviewer’s feedback. The sentence has been revised for clarity:
"The aerobic oxidation of hydrocarbons in *Ca. Aerovita* expands its potential carbon sources by enabling the conversion of hydrocarbons into alcohols, which can then be metabolized through alcohol metabolism pathways inherited from ancestral *Ca. Aerarchaeales*.**

Reviewer #2 (Remarks to the Author):

RE: ‘Aerobic hydrocarbon oxidation in a novel archaeal lineage’

This manuscript proposes aerobic hydrocarbon oxidation by Candidatus 'Aerovita' of Syntropharchaeota based on analysis of metagenome-assembled genomes (MAGs). The authors obtained three MAGs of this new genus and identified copper-containing membrane monooxygenase (CuMMO) genes, suggesting a hydrocarbon oxidation trait for this clade. Based on the presence of oxygen-dependent enzymes, the authors propose a shift from anaerobic to aerobic lifestyle via horizontal gene transfer between archaea and bacteria. The novelty lies in the fact that no archaea have yet been reported to oxidize hydrocarbons aerobically, although Syntropharchaeota contains ACR genes involved in anaerobic alkane oxidation (propane, butane, and long-chain alkanes). Together, aerobic alkane oxidation has only been reported in bacteria, and this study proposes to extend this trait to archaea.

Major comments:

The term "hydrocarbon monooxygenase" (HMO) should be more specific. CuMMO typically refers to short-chain hydrocarbon monooxygenase, while long-chain hydrocarbon monooxygenase contains a diiron center.

R10. We appreciate the reviewer’s feedback. In response, we have clarified in lines 122–125 of our manuscript that CuMMOs are known for their substrate promiscuity, capable of oxidizing a wide range of hydrocarbons, including both short- and long-chain alkanes (C2-C24). Therefore, we have chosen to use the more general term *hydrocarbon monooxygenase* to reflect the broad substrate specificity of these enzymes across various hydrocarbons, rather than limiting it to a specific chain length. We respectfully believe this terminology more accurately captures the diversity of potential substrates that CuMMOs can act upon.

CuMMO is known to be promiscuous in terms of substrate. Downstream pathways, rather than the enzyme itself, often determine the physiological roles of CuMMO. Phylogenetic evidence only partially supports the enzyme's physiological function. As the authors noted, many phylogenetically divergent phyla contain CuMMO, but their functions cannot be definitively classified as HMOs without further evidence (e.g., gene expression analysis).

Although alcohol and aldehyde dehydrogenase genes were reported in the MAGs, these are present in multiple copies and can be involved in various metabolic pathways of general heterotrophs. In addition, in the manuscript, it is described that fatty acid utilization is LCA trait of Syntropharchaeota, meaning not related to presence of CuMMO.

R11. Regarding the presence of alcohol and aldehyde dehydrogenases in the MAGs, we acknowledge the reviewer's point that these enzymes are found in multiple copies and are involved in various metabolic pathways in general heterotrophs. However, we respectfully maintain that the presence of these genes does not rule out their potential involvement in the aerobic hydrocarbon oxidation process.

In response to the comment on fatty acid metabolism being a trait of *Syntropharchaeia*, we agree that this pathway is part of the lineage's ancestral metabolism. However, we would like to respectfully maintain that the acquisition of CuMMO genes in *Ca. Aerovita* would have integrated into and augmented the existing fatty acid metabolism pathway.

It would be beneficial to determine if these genes are localized in the pseudoperiplasm, as in other CuMMO cases (e.g., AMO, PMO).

R12. The CuMMO operon in *Ca. Aerovita* encodes the subunit *hmoC*, which is essential for the transfer of electrons from the menaquinone pool to the CuMMO complex, enabling the catalytic oxidation of hydrocarbons. Based on this, we infer that the CuMMO complex is likely membrane-bound, similar to other CuMMO-containing enzymes such as AMO and PMO.

Based solely on genomic analysis, without additional supporting evidence, it is challenging to claim this novel trait, which is a key aspect of the manuscript.

R13. We appreciate the reviewer's comment and acknowledge that genomic data alone may not provide definitive proof of the observed metabolic traits. However, genomic analysis, particularly the identification of a novel lineage containing horizontally transferred genes such as the CuMMO operon and complex IV, provides strong evidence for the potential of oxygen-respiring metabolism in *Ca. Aerovita*. These genes are well-characterized in other organisms and are involved exclusively in aerobic respiration. The data we present represents a significant step toward understanding the potential ecological roles of this novel archaeal lineage, warranting further research and investigation.

While present in various environments as any microbe, the abundance of this archaeal lineage appears to be low, and its role as alkane oxidizers in ecosystems should be carefully considered. General information.

R14. We agree that the role of this archaeal lineage as alkane oxidizers in ecosystems should be carefully considered. We do not claim that this lineage is a dominant alkane oxidizer in ecosystems, however, its widespread presence across diverse geographical locations suggests its potential global significance.

Specific comments:

Line 122: If there is evidence of CuMMO involvement in long-chain alkane oxidation, it should be described in detail and discussed. I don't know about that.

R15. As noted in our previous response (R10), we have referenced evidence of CuMMO's involvement in long-chain alkane oxidation.

Lines 114-116: Consider whether three MAGs provide sufficient evidence for the claim made.

**R16. We appreciate the reviewer's comment. The sentence has been amended to:
"The MAGs from g__CG-2 and f__YL-1 did not encode genes for the CuMMO complexes."**

Lines 145-147: Clarify whether the conservation is specific to HMO or applies to all CuMMO. I think it is conserved in all CuMMO.

R17. Thank you for this suggestion. We have already discussed the conservation of residues across all CuMMOs in Lines 140–145. In Lines 145–147, we further emphasize that the phylogenetic analysis supports our conclusion that the CuMMO complexes are likely functionally conserved hydrocarbon monooxygenases.

Lines 178-179: Elaborate on potential electron donors for CO₂ fixation.

R18. CO₂ reduction via carbon-monoxide dehydrogenase is indeed a well-established process that requires oxidized ferredoxin as an electron donor. We will include this pathway in our metabolic cartoon in Figure 4, as suggested, to further clarify this metabolic step.

Lines 184-187: Clarify the relationship between incomplete electron transport and the presence of rTCA.

R19. Apologies for the confusion. The sentence was intended to highlight the differences between the metabolic pathways of f__YL-1 and f__CG-2. To clarify the comparison, we will revise the sentence to:

"In contrast to f__CG-2, the f__YL-1 MAGs do not encode a complete electron transport chain or TCA cycle."

Lines 202-204: Beta oxidation pathway enzymes themselves may not be sensitive to oxygen. If beta oxidation is a trait of the last common ancestor of Syntropharchaeota, it is not unique to the CuMMO MAGs, potentially weakening the connection between CuMMO and fatty acid oxidation.

R20. We acknowledge the reviewer's point regarding the beta-oxidation pathway enzymes. We have already addressed the connection between CuMMO and fatty acid oxidation in R14. To further clarify, we emphasize that while beta-oxidation may not be unique to the CuMMO MAGs, the integration of CuMMO within the broader fatty acid metabolism in *Ca. Aerovita* adds a compelling dimension to its potential role in aerobic hydrocarbon oxidation.

REVIEWER COMMENTS

Reviewer #1 (Remarks to the Author):

Comment 1.1

I appreciate the authors' responses and agree that revising the term "aerobic lifestyle" to "aerobic respiration" improves clarity. I also concur with the conclusion that these archaea are very likely capable of oxygen-dependent hydrocarbon degradation—a novel and intriguing finding—and that they can respire oxygen. However, I still find issues in the reconstruction and interpretation of these archaea's metabolism that need to be addressed.

Response 1.1

We are pleased that Reviewer 1 agrees with our revised framing of the archaeal lineage as capable of aerobic respiration rather than exhibiting an "aerobic lifestyle." We fully agree that this wording more accurately reflects our findings and avoids overextending interpretations beyond the available data.

Comment 1.2

As noted in my previous comments, the aerobic metabolism described for this archaeal lineage is atypical and warrants explicit clarification. Without sufficient explanation, the proposed electron transport system appears confusing, especially in comparison to typical aerobic respiration pathways found in model bacteria. According to the authors' data, a key respiratory complex—complex I (NADH dehydrogenase)—lacks important subunits (e.g., NuoEFG; see Extended Data Fig. 5). The authors infer that this truncated complex instead oxidizes ferredoxin. Based on their annotations (e.g., Fig. 4), this partial complex I and FixABCX are the only candidate electron carrier-oxidizing enzymes that could supply electrons into the quinone pool for aerobic respiration. This implies that ferredoxin is centrally involved in the oxygen respiratory chain of these archaea—an unusual feature, given ferredoxin's general oxygen sensitivity.

In my previous comments, I noted that this observation is at odds with the aerobic lifestyle the authors attribute to these organisms. The authors disagree, arguing that the presence of ferredoxin-dependent enzymes does not imply facultative anaerobiosis or microaerophily, citing examples of "obligate aerobe" that use ferredoxin (Response 1.3). However, the cited examples are not broadly representative. Specifically:

1. *S. solfataricus* and *H. salinarum* can grow anaerobically via anaerobic respiration (see 10.1099/ijsem.0.002665 and 10.3389/fmicb.2022.811432/full).
2. These archaea inhabit extreme environments (high temperature or high salinity) where oxygen solubility is low—*Sulfolobus* is hyperthermophilic, *Halobacterium* and *N. magadii* are halophilic (see 10.1038/npg.els.0000394).
3. In some cases, ferredoxin is used in conjunction with protective strategies against oxygen, such as cell-coating with alginate, as in *Azotobacter vinelandii*, which also uses ferredoxin for nitrogen fixation (see 10.1128/AEM.66.9.4037-4044.2000). The use of FixABCX in these taxa is also linked to low-oxygen adaptations like nitrogenase function.

In other words, these examples of "obligate aerobe" are not actually paradigms of high-

oxygen adaptation, and they share ecological or physiological features that allow the coexistence of oxygen sensitivity and respiration. Critically, none of the environments from which the authors' MAGs are derived appear to share such extreme or microoxic characteristics. This undermines the generalizability of the cited examples and emphasizes the uniqueness of the finding.

Indeed, the study the authors themselves cite (Yan et al., Sci Rep 2016) notes that oxygen-sensitive OFORs are “currently only observed in aerobic organisms living in extreme conditions, such as high temperature or high salt concentrations.” The present MAGs are neither. Therefore, rather than rebutting the concern, the cited cases actually reinforce that the co-occurrence of ferredoxin and oxygen respiration outside of extreme environments is uncommon—and potentially notable. I strongly recommend that the authors frame this feature as a novel metabolic configuration that deserves attention (e.g., thermophilic, halophilic, and now this lineage).

In Response 1.3, the authors refer to KEGG data showing ferredoxin-linked enzymes in other aerobic archaea. Could the authors further examine and report on the ecological distribution of these archaea to evaluate whether they also occupy low-O₂ or extreme environments? My guess is that most are halophiles or hyperthermophiles. Again, this context would highlight what makes their current archaeal group distinctive.

Response 1.2

We thank Reviewer 1 for their patience and thoughtful, in-depth feedback. We would like to clarify that it was not our intention to suggest that our archaeal lineage operates in high-oxygen environments. We acknowledge that our previous response may have unintentionally conveyed this impression. In our initial response, we cited several studies that used the presence of cytochrome c oxidase (Cox) as an indicator of an aerobic lifestyle. However, we recognize that such inferences are insufficient without complementary physiological or *in situ* activity data.

In our manuscript, we referenced *Ca. Angelarchaeales*, a novel Thermoplasmata order with a divergent CuMMO predicted to be an “obligate aerobe” and to mediate ammonia oxidation. Like ammonia-oxidizing Nitrososphaerales (AOA), it encodes blue copper proteins hypothesized to mediate the downstream pathway of ammonia oxidation. Like our archaeon, it encodes a truncated complex I lacking NADH-binding subunits (E, F, G) that likely uses reduced ferredoxin, succinate/fumarate dehydrogenase, a cytochrome b-like complex III, cytochrome c oxidase, and a FixABCX complex.

Although detected in oxygenated habitats, *Ca. Angelarchaeales* reaches its highest abundance at 20–40 cm soil depth, conditions typically microaerophilic, and is found alongside microaerophile-adapted AOA¹, suggesting low-oxygen activity despite earlier “obligate aerobe” designations made without direct oxygen measurements. Given the similar metabolic features, this initially led us to propose an aerobic lifestyle for our archaeon. However, your feedback and our reassessment of the literature indicate that aerobic growth remains unsubstantiated without further physiological assays. We have therefore limited our discussion to the potential aerobic respiratory metabolism of our lineage.

One of the key novelties you highlighted was that the “co-occurrence of ferredoxin and oxygen respiration outside of extreme environments is uncommon” and should be emphasized. We acknowledge that the metabolism we describe is indeed different from the extremophilic aerobic archaea. While ferredoxins are oxygen sensitive, particularly those with [4Fe-4S] clusters, the [2Fe-2S] ferredoxins are generally more tolerant of oxygen exposure². We identified [2Fe-2S] ferredoxins exclusively in CM-1, CG-1, and CG-2, all of which encode complex IV. Notably, the [2Fe-2S] ferredoxin was absent in YL-1 and YL-2, and other members of the Syntropharchaea.

We have now expanded the relevant finding in the manuscript in Ln 189-193:

“Genes encoding [2Fe-2S] ferredoxin, known to be more oxygen-tolerant⁵³ were identified exclusively in members of f__CG-2. Although the specific redox partner for this ferredoxin could not be determined, its absence in other members of Syntropharchaea suggests a potential role in aerobic respiration, and an electron donor source for the HMO complex.”

Comment 1.3

In addition, a comparison with canonical aerobic hydrocarbon-degrading bacteria would strengthen the manuscript. These bacteria typically employ a very different metabolic setup (e.g., full complex I, NADH as electron donor, etc.). A few sentences contrasting these bacteria with the archaea studied here would help readers appreciate the unusual nature of this archaeal metabolism.

Response 1.3

We are happy to incorporate a section discussing the novelty of this metabolic configuration in the discussion Ln 312-326:

“This observation raises important questions about the mechanisms of CuMMO activation across diverse lineages. In aerobic methanotrophic and alkanotrophic bacteria, pMMO/HMO activity is stimulated by reduced quinones⁶⁵, which are generated via NADH oxidation and quinone reduction by complex I⁶⁶. In contrast, Nitrososphaerales AOA and *Ca.* ‘Angelarchaeales’ within Thermoproteota and Thermoplasmata, respectively are thought to generate reducing equivalents for AMO via alternative mechanisms. These include hydroxylamine oxidation coupled with quinone reduction by blue copper proteins^{48,67}, or ferredoxin oxidation via a truncated complex I that lacks the NADH-binding domain⁶⁸. In *Ca.* ‘Angelarchaeales’, additional ferredoxin-oxidizing systems have been observed, such as a FixABCX complex that couples the NADH oxidation to the simultaneous reduction of ferredoxin and quinone⁴⁸. In *Ca.* ‘Aerovita’, we identified a truncated complex I, a FixABCX complex, and an oxygen-tolerant [2Fe-2S] ferredoxin, suggesting a distinctive ferredoxin-dependent metabolic configuration for CuMMO activation in archaea. These findings point to an underappreciated diversity in archaeal respiratory strategies, likely shaped by repeated HGT events.”

Comment 1.4

Regarding Cox as a marker for aerobic respiration, the authors state it is associated with

respiration at higher oxygen concentrations (Response 1.2). However, recent evidence suggests otherwise—an obligately anaerobic archaeon and other anaerobic bacteria also encode Cox (see 10.1101/2025.02.26.640444), indicating that some Cox types may function under low-O₂ conditions. This weakens its utility as a definitive marker of “aerobic lifestyle.”

Response 1.4

We agree with your assessment. Please see Response 1.

Comment 1.5

Similarly, the authors suggest that the absence of the Wood–Ljungdahl pathway supports an aerobic lifestyle. This is not a valid inference; many anaerobes lack this pathway, and its absence does not equate to aerobic metabolism.

Response 1.5

We have removed all mention of this inference from our manuscript.

Comment 1.6

Finally, regarding Fig. 4: the depiction of electron-bifurcating butyryl-CoA dehydrogenase appears to have the directionality reversed. NAD⁺ should be reduced and ferredoxin oxidized during beta-oxidation, not the other way around.

Response 1.6

We thank Reviewer 1 for spotting this error. We have changed Figure 4.

Reviewer #2 (Remarks to the Author):

Comment 2.1

General Assessment

The revised manuscript is significantly improved. However, inevitably, the novelty of the discovery is diminished by the fact that the functional claims are based solely on genomic analysis of microorganisms that are distantly related to previously known HMO (CuMMO) lineages, and by their low abundance in the environment (despite being widespread). Phylogenetic and similarity analyses of HMO genes may not be particularly valuable for such distantly related CuMMO microorganisms. That said, given the inherent limitations of MAGs, the manuscript’s claims are reasonably supported by the genomic data. The overall narrative and logic of the manuscript remain intact and well-written. I agree with the

suggestion that future work should focus on increasing genomic representation and cultivating members of this lineage to confirm their metabolism and ecological importance.

Response 2.1

We thank Reviewer 2 for taking the time to reassess our manuscript and commenting that our manuscript is significantly improved.

Comment 2.2

Comments on Authors' Responses

Response 2.1 and 2.6

Regarding the statement: “Therefore, we have chosen to use the more general term hydrocarbon monooxygenase to reflect the broad substrate specificity of these enzymes across various hydrocarbons, rather than limiting it to a specific chain length.”

Upon reviewing the four cited references, the evidence indicates that CuMMO/HMO enzymes are limited to short-chain (C2–C4) gaseous hydrocarbons. Please review these references again, as they do not support activity beyond C4 alkanes.

Response 2.1

We thank the reviewer for pointing out our mistake. Our sentence indeed describes both the particulate and soluble hydrocarbon monooxygenase. The HMO characterized in *Mycobacterium* by Coleman et al. showed activity on alkanes in the C2–C4 range, as well as some alkenes and halogenated hydrocarbons [1].

1. Coleman, N. V. *et al.* Hydrocarbon monooxygenase in *Mycobacterium*: recombinant expression of a member of the ammonia monooxygenase superfamily. *The ISME journal* **6**, 171-182 (2012).

We have thus revised this sentence in Ln 122-125:

“Characterized HMOs have been shown to act specifically on short-chain hydrocarbons, including alkanes (C2–C4), alkenes, and chlorinated alkanes³³. Thus, while the exact substrate(s) of the g__CM1 HMO proteins remain unknown, they are likely involved in the oxidation of short-chain hydrocarbons.”

Comment 2.2

Response 2.2

While the process of elimination suggests a close relationship to HMO microorganisms, with a low likelihood of AMO or PMO involvement, other functional possibilities remain. Many CuMMO enzymes, such as pxmABC, have unknown functions and are widespread, as discussed in the literature. For example, pxmABC is prevalent but its function remains unclear, and other CuMMOs in Alpha-, Beta-, and Gamma-proteobacteria are found near the

base of Fig. 1 (the first ref). Recent studies have also suggested acetone monooxygenase activity from PMO-like gene clusters (the second one).

Response 2.2

Thank you for the helpful comment. While our phylogenetic analysis and gene context suggest a closer relationship to HMO-type CuMMOs, we agree that other functions are possible. As noted, pxmABC-like CuMMOs are widespread, yet their substrates remain unknown. Recent studies also suggest PMO-like clusters may oxidize acetone; however, we did not identify GMC oxidoreductases, which are required to process acetol, the product of acetone oxidation. Given the lack of downstream genes for ammonia, methane, and acetone oxidation pathways, and based on current evidence, we propose hydrocarbon oxidation as the most likely function.

We have revised the downstream analysis pathway section to better address the broad substrate specificity of CuMMOs in Ln 158-173:

“While the substrate range of CuMMO enzymes can be broad, with pMMOs capable of oxidizing methane, short-chain alkanes (C₂–C₅), alkenes (C₂–C₄)^{43,44}, and acetone⁴⁵, and AMOs known to act on ammonia, as well as alkanes (≤C₈) and alkenes (≤C₅)⁴⁶, the actual substrate specificity is often determined by downstream metabolic pathways. Key genes essential for bacterial aerobic methane oxidation, including methanol dehydrogenase (*mdh*), methylene tetrahydrofolate dehydrogenase (*folD*), and methylene-tetrahydromethanopterin dehydrogenase (*mtdB*), were absent in g__CM-1, suggesting methane is unlikely to be its primary substrate. No genes encoding glucose-methanol-choline oxidoreductase, typically required for processing acetol, the product of acetone oxidation were identified. Similarly, we did not detect hydroxylamine dehydrogenase (*hao*) genes characteristic of ammonia-oxidizing bacteria⁴⁷, indicating bacterial ammonia oxidation is improbable. In ammonia-oxidizing archaeal (AOA) lineages *Ca.* ‘Angelarchaeales’ and ‘Nitrososphaerales’, conserved plastocyanin-like proteins putatively replace *hao*⁴⁸, though their function remains to be determined. While plastocyanin-like proteins were identified in g__CM-1 (**Supplementary Table 4**), they showed higher sequence similarity to non-AOA bacterial and archaeal lineages, suggesting they are not involved in ammonia oxidation.”

Comment 2.3

Response 2.3

I apologize for the earlier lack of clarity. I intended to highlight that alcohol dehydrogenase (ADH) is usually localized outside the cell membrane. Please check the localization of ADH,

not CuMMO. If all ADHs are cytoplasmic, you can propose the presence of transporters for alcohol intermediates.

Response 2.3

Thank you for the clarification. Using both SignalP 6.0 and PSORTb, all predicted ADH proteins in our dataset were consistently assigned as cytoplasmic. This localization suggests that any alcohol substrates processed by these enzymes must be intracellular.

A search of the Transporter Classification Database (TCDB) revealed no transporters with definitive homology or annotation indicating specificity for the uptake of small alcohols (e.g., ethanol, propanol, butanol). This is consistent with current literature, where no dedicated alcohol transporters have been functionally characterized in archaea or bacteria to date.

Given that small, uncharged alcohols (C1–C4) are known to permeate biological membranes via passive diffusion, the absence of identifiable transporters is expected and does not preclude alcohol uptake. While we did identify several putative ABC transport systems, their substrate specificity remains unclear, and any functional inference regarding alcohol uptake would be speculative.

Comment 2.4

Response 2.9

My previous comment referred not to the direct electron donor, but to the ultimate source of energy (e.g., phototrophy or lithotrophy) for CO₂ fixation. Please clarify whether CO₂ fixation is linked to autotrophic growth in these organisms.

Response 2.4

Thank you for clarifying your question. We believe our f__YL-1 lineage is capable of lithoautotrophy via hydrogen as an electron donor source. We have elaborated on this process in Figure 4 and in the following section in Ln 199-227:

“In contrast to f__CG-2, the f__YL-1 MAGs encoded genes for the Wood-Ljungdahl pathway (WLP) but lacked the tetrahydromethanopterin S-methyltransferase (Mtr) and MCR/ACR complexes (Figure 3 and Figure 4). Notably, the *mtrH* gene which is crucial for transfer of methyl groups⁵⁴ was identified, linking methyl-CoM and the methyl-branch of the Wood-Ljungdahl pathway. Thus, members of f__YL-1 are unlikely to oxidise methane or hydrocarbons, instead using the Wood-Ljungdahl pathway to fix carbon dioxide. The presence of the WLP genes alongside group 3b and group 4g [NiFe] hydrogenases (Figure 4) suggests the capacity for lithoautotrophic growth using H₂ as an electron donor. The group 3b hydrogenase mediates the reversible oxidation of H₂ to the reduction of NADPH, and potentially to the reduction of an unknown electron carrier, which has not been characterised *in vivo*⁵⁵. In both the YL-1 and YL-2 MAGs, the group 3b gene clusters were co-localised

with a heterodisulfide reductase A (*hdrA*) subunit (**Supplementary Table 4**). Electrons from the group 3b hydrogenase could be transferred to HdrA which is involved in flavin-based electron bifurcation, mediating the reduction of ferredoxin and sending electrons to *hdrBC*, which mediates the reduction of CoB-S-S-CoM⁵⁶. The group 4g hydrogenase, though not yet biochemically characterised⁵⁷, encodes an antiporter-like membrane subunit that suggests sodium/proton translocation. In *f__YL-1*, the group 4g hydrogenase gene clusters were found adjacent to a coenzyme F₄₂₀-reducing [NiFe]-hydrogenase B subunit (**Supplementary Table 4**), which mediates F₄₂₀ reduction⁵⁸, suggesting H₂ oxidation could be coupled to the reduction of F₄₂₀ in contrast to ferredoxin seen typically in other group 4 NiFe hydrogenases in methanogens⁵⁹. Additionally, soluble F₄₂₀H₂ dehydrogenase subunit F genes were identified, potentially enabling reversible electron transfer between F₄₂₀H₂ and ferredoxin. Like *f__CG-2*, *f__YL-1* also possess genetic potential to metabolize long-chain fatty acids, propionate, and alcohols, indicating the capacity for mixotrophic growth on both inorganic and organic carbon sources. In contrast to *f__CG-2*, the *f__YL-1* MAGs do not encode a complete electron transport chain or TCA cycle. Instead, they likely use an incomplete rTCA cycle (Extended Figure 5) to funnel acetyl-CoA from the Wood-Ljungdahl pathway and other heterotrophic pathways into the other universal precursors of anabolism (e.g., pyruvate, phosphoenolpyruvate, oxaloacetate, and 2-oxoglutarate)⁶⁰.”

Comment 2.5

Additional Comments

Line 67 and elsewhere: The term “mixotrophic” is unclear. Please define it at first use.

Response 2.5

We are happy to revise the sentence in Ln 64-68:

“Comparative genomics and metabolic reconstruction of six metagenome assembled genomes reveal the acquisition of the HMO complex and electron transport complexes within a novel genus, supporting their potential transition from a primarily anaerobic mixotrophic metabolism that includes organic matter degradation and CO₂ fixation towards an aerobic hydrocarbon-oxidising metabolism.”

Comment 2.6

Figure 1: Please add the order label ‘JACQPP01’ for clarity.

Response 2.6

We have relabelled Figure 1 for clarity.

Comment 2.7

Extended Figure 2: The figure is almost illegible due to the small font size. Please increase the font size for readability. The (deep) branching of the HMO should also be more clearly visualized.

Response 2.7

We have changed Extended Figure 2 to improve legibility.

Comment 2.8

Line 257: Please clarify whether the ANI% refers to the 16S rRNA gene sequence.

Response 2.8

This has been clarified in Ln 281-283:

“To investigate the distribution of *Ca.* ‘Aerarchaeales’, we searched for closely related 16S rRNA gene sequences ($\geq 90\%$ identity) using the SILVA 138.1 release, revealing five sequences from terrestrial and marine ecosystems (Figure 6).”